# Sample-Efficient Linear Representation Learning from Non-IID Non-Isotropic Data

**Thomas T.C.K. Zhang**
University of Pennsylvania
Philadelphia, PA
`ttz2@seas.upenn.edu`

**Leonardo F. Toso**
Columbia University
New York, NY
`lt2879@columbia.edu`

**James Anderson**
Columbia University
New York, NY
`james.anderson@columbia.edu`

**Nikolai Matni**
University of Pennsylvania
Philadelphia, PA
`nmatni@seas.upenn.edu`

## Abstract

A powerful concept behind much of the recent progress in machine learning is the extraction of common features across data from heterogeneous sources or tasks. Intuitively, using all of one's data to learn a common representation function benefits both computational effort and statistical generalization by leaving a smaller number of parameters to fine-tune on a given task. Toward theoretically grounding these merits, we propose a general setting of recovering linear operators $M$ from noisy vector measurements $y = Mx + w$, where the covariates $x$ may be both non-i.i.d. and non-isotropic. We demonstrate that existing isotropy-agnostic meta-learning approaches incur biases on the representation update, which causes the scaling of the noise terms to lose favorable dependence on the number of source tasks. This in turn can cause the sample complexity of representation learning to be bottlenecked by the single-task data size. We introduce an adaptation, `De-bias & Feature-Whiten` (`DFW`), of the popular alternating minimization-descent (AMD) scheme proposed in Collins et al., (2021) (Collins et al., 2021), and establish linear convergence to the optimal representation with noise level scaling down with the *total* source data size. This leads to generalization bounds on the same order as an oracle empirical risk minimizer. We verify the vital importance of `DFW` on various numerical simulations. In particular, we show that vanilla alternating-minimization descent fails catastrophically even for iid, but mildly non-isotropic data. Our analysis unifies and generalizes prior work, and provides a flexible framework for a wider range of applications, such as in controls and dynamical systems.

## 1 Introduction

A unifying paradigm belying recent exciting progress in machine learning is learning a common feature space or *representation* for downstream tasks from heterogeneous sources. This forms the core of fields such as meta-learning, transfer learning, and federated learning. A shared theme across these fields is the scarcity of data for a specific task out of many, such that designing individual models for each task is both computationally and statistically inefficient, impractical, or impossible. Under the assumption that these tasks are similar in some way, a natural alternative approach is to use data across many tasks to learn a common component, such that fine-tuning to a given task involves fitting a much smaller model that acts on the common component. Over the last few years, significant attention has been given to framing this problem setting theoretically, providing provable benefits of learning over multiple tasks in the context of linear regression (Bullins et al., 2019; Du et al., 2020; Tripuraneni et al., 2021; Collins et al., 2021; Thekumparampil et al., 2021; Saunshi et al., 2021) and in identification/control of linear dynamical systems (Modi et al., 2021; Chen et al., 2023; Zhang et al., 2023). These works study the problem of *linear representation learning*, where the data for each task is generated noisily from an unknown shared latent subspace, and the goal is to efficiently recover a representation of the latent space $\hat{\Phi}$ from data across different task distributions.

For example, in the linear regression setting, one may have data of the form

$$y_i^{(t)} = {\theta^{(t)}}^\top \Phi x_i^{(t)} + \text{noise}, \quad y_i^{(t)} \in \mathbb{R}, x_i^{(t)} \in \mathbb{R}^{d_x}, \Phi \in \mathbb{R}^{r \times d_x},$$

with $i = 1, \dots, N$ iid data points from $t = 1, \dots, T$ task distributions. Since the representation $\Phi$ is shared across all tasks, one may expect the generalization error of an approximate representation $\hat{\Phi}$ fit on $TN$ data points to scale as $\frac{d_x r}{TN}$, where $d_x r$ is the number of parameters determining the representation. This is indeed the flavor of statistical guarantees from prior work (Du et al., 2020; Tripuraneni et al., 2021; Thekumparampil et al., 2021; Zhang et al., 2023), which concretely demonstrates the benefit of using data across different tasks.

However, existing work, especially beyond the scalar measurement setting, is limited in one or more important components of their analysis. For example, it is common to assume that the covariates $x_i^{(t)}$ are isotropic across all tasks. Furthermore, statistical analyses often assume access to an empirical risk minimizer, even though the linear representation learning problem is non-convex and ill-posed (Maurer et al., 2016; Du et al., 2020; Zhang et al., 2023). Our paper addresses these problems under a unified framework of *linear operator recovery*, i.e. recovering linear operators $M \in \mathbb{R}^{d_y \times d_x}$ from (noisy) vector measurements $y = Mx + w$, where the covariates $x$ may not be independent or isotropic. This setting subsumes the scalar measurement setting, and encompasses many fundamental control and dynamical systems problems, such as linear system identification and imitation learning. In particular, the data in these settings are incompatible with the common distributional assumptions (e.g., independence, isotropy) made in prior work.

**Contributions:** Toward this end, our main contributions are as follows:

- We demonstrate that naive implementation of local methods for linear representation learning fail catastrophically even when the data is iid but mildly non-isotropic. We identify the source of the failure as interaction between terms incurring biases in the representation gradient, which do not scale down with the number of tasks.
- We address these issues by introducing two practical algorithmic adjustments, `De-bias & Feature-Whiten` (`DFW`), which provably mitigate the identified issues. We then show that `DFW` is necessary for gradient-based methods to benefit from the total size of the source dataset.
- We numerically show our theoretical guarantees are predictive of the efficacy of our proposed algorithm, and of the key importance of individual aspects of our algorithmic framework.

Our main result can be summarized by the following descent guarantee for our proposed algorithm.

**Theorem 1.1 (main result, informal)** *Let $\hat{\Phi}$ be the current estimate of the representation, and $\Phi_\star$ the optimal representation. Running one iteration of DFW yields the following improvement*

$$\text{dist}(\hat{\Phi}_+, \Phi_\star) \le \rho \cdot \text{dist}(\hat{\Phi}, \Phi_\star) + \frac{C}{\sqrt{\# \text{ tasks} \times \# \text{ data per task}}}, \quad \rho \in (0, 1), \ C > 0.$$

Critically, the second term of the right hand side scales jointly in the number of tasks and datapoints per task, whereas naively implementing other methods may be bottlenecked by a term that scales solely with the amount of data for a single task, which leads to suboptimal sample-efficiency.

## 1.1 RELATED WORK

**Multi-task linear regression:** Directly related to our work are results demonstrating the benefits of multi-task learning for linear regression (Maurer et al., 2016; Bullins et al., 2019; Du et al., 2020; Tripuraneni et al., 2021; Collins et al., 2021; Thekumparampil et al., 2021), under the assumption of a shared but unknown linear feature representation. In particular, our proposed algorithm is adapted from the alternating optimization scheme independently in Nayer and Vaswani (2022) and Collins et al. (2021), and extends these results to the vector measurement setting and introduces algorithmic modifications to extend its applicability to non-iid and non-isotropic covariates. We also highlight that in the isotropic linear regression setting, Thekumparampil et al. (2021) provide an alternating minimization scheme that results in near minimax-optimal representation learning. However, the representation update step simultaneously accesses data across tasks, which we avoid in this work due to motivating applications, e.g. distributed learning, that impose locality or data privacy constraints.

**Meta/multi-task RL:** There is a wealth of literature in reinforcement learning that seeks empirically to solve different tasks with shared parameters (Teh et al., 2017; Hessel et al., 2018; Singh et al., 2020; Deisenroth et al., 2014). In parallel, there is a body of theoretical work which studies the sample efficiency of representation learning for RL (Lu et al., 2021; Cheng et al., 2022; Maurer et al., 2015). This line of work considers MDP settings, and thus the specific results are often stated with incompatible assumptions (such as bounded states/cost functions and discrete action spaces), and are suboptimal when instantiated in our setting.

**System identification and control:** Multi-task learning has gained recent attention in controls, e.g. for adaptive control over similar dynamics (Harrison et al., 2018; Richards et al., 2021; Shi et al., 2021; Muthirayan et al., 2022), imitation learning for linear systems (Zhang et al., 2023; Guo et al., 2023), and notably linear system identification (Li et al., 2022; Wang et al., 2022; Xin et al., 2023; Modi et al., 2021; Faradonbeh and Modi, 2022; Chen et al., 2023). In many of these works (Li et al., 2022; Wang et al., 2022; Xin et al., 2023), task similarity is quantified by a generic norm closeness of the dynamics matrices, and thus the benefit of multiple tasks extends only to a radius around optimality. Under the existence of a shared representation, our work provides an efficient algorithm and statistical analysis to establish convergence to per-task optimality.

## 2 PROBLEM FORMULATION

**Notation:** the Euclidean norm of a vector $x$ is denoted $\|x\|$. The spectral and Frobenius norms of a matrix $A$ are denoted $\|A\|$ and $\|A\|_F$, respectively. For symmetric matrices $A, B$, $A \preccurlyeq B$ denotes $B - A$ is positive semidefinite. The largest/smallest singular and eigenvalues of a matrix $A$ are denoted $\sigma_{\max}(A)$, $\sigma_{\min}(A)$, and $\lambda_{\max}(A)$, $\lambda_{\min}(A)$, respectively. The condition number of a matrix $A$ is denoted $\kappa(A) := \sigma_{\max}(A)/\sigma_{\min}(A)$. Define the indexing shorthand $[n] := \{1, \ldots, n\}$. We use big-O notation $\mathcal{O}(\cdot)$, $\Theta(\cdot)$, $\Omega(\cdot)$ to omit universal numerical factors, and $\tilde{\mathcal{O}}(\cdot)$, $\tilde{\Theta}(\cdot)$, $\tilde{\Omega}(\cdot)$ to additionally omit polylog factors in the argument.

**Regression Model.** Let a covariate sequence (also denoted a *trajectory*) be an indexed set $\{x_i\}_{i\geq 1} \subset \mathbb{R}^{d_x}$. We denote a distribution $\mathbb{P}_x$ over covariate sequences, which we assume to have bounded second moments for all $i \geq 1$, i.e. $\mathbb{E}\left[x_i x_i^\top\right]$ is finite for all $i \geq 1$. Defining the filtration $\{\mathcal{F}_i\}_{i\geq 0}$ where $\mathcal{F}_i := \sigma(\{x_k\}_{k=1}^{i+1}, \{w_k\}_{k=1}^{i})$ is the $\sigma$-algebra generated by the covariates up to $i+1$ and noise up to $i$, we assume that $\{w_i\}_{i\geq 1}$ is a $\sigma_w^2$-subgaussian martingale difference sequence (MDS):

$$\mathbb{E}\left[v^\top w_i \mid \mathcal{F}_{i-1}\right] = 0, \quad \mathbb{E}\left[\exp\left(\lambda v^\top w_i\right) \mid \mathcal{F}_{i-1}\right] \leq \exp\left(\lambda^2 \|v\|^2 \sigma_w^2\right) \text{ a.s. } \forall \lambda \in \mathbb{R}, v \in \mathbb{R}^{d_y}.$$

Assuming a ground truth operator $M_\star \in \mathbb{R}^{d_y \times d_x}$, our observation model is given by

$$y_i = M_\star x_i + w_i, \quad i \geq 1,$$

for $y_i$ the *labels*, and $w_i$ the *label noise*. We further define $\Sigma_{x,N} := \frac{1}{N}\sum_{i=1}^N \mathbb{E}[x_i x_i^\top]$. When the marginal distributions of $x_i$, $i \geq 1$ are identical, we denote $\Sigma_x \equiv \Sigma_{x,N}$.

**Multi-Task Operator Recovery.** We consider the following instantiation of the above linear operator regression model over multiple tasks. In particular, we consider heterogeneous data $\{(x_i^{(t)}, y_i^{(t)})\}_{i=1,t=1}^{N,T}$, consisting of trajectories of length $N$, generated independently across $t = 1, \ldots, T$ task distributions. For notational convenience, we assume that the length of trajectories $N$ is the same across training tasks. For each task $t$, the observation model is

$$y_i^{(t)} = M_\star^{(t)} x_i^{(t)} + w_i^{(t)}, \tag{1}$$

where $M_\star^{(t)} = F_\star^{(t)} \Phi_\star$ admits a decomposition into a ground-truth representation $\Phi_\star \in \mathbb{R}^{r \times d_x}$ common across all tasks $t \in [T]$ and a task-specific weight matrix $F_\star^{(t)} \in \mathbb{R}^{d_y \times r}$, $r \leq d_x$. We denote the joint distribution over covariates and observations $\{x_i^{(t)}, y_i^{(t)}\}_{i\geq 1}$ by $\mathbb{P}_{x,y}^{(t)}$. We assume that the representation $\Phi_\star$ is normalized to have orthonormal rows to prevent boundedness issues, since $F_\star^{(t)\prime} = F_\star^{(t)} Q^{-1}$, $\Phi_\star' = Q\Phi_\star$ are valid decompositions for any invertible $Q \in \mathbb{R}^{r \times r}$. To measure closeness of an approximate representation $\hat{\Phi}$ to optimality, we define a subspace metric.

**Definition 2.1 (Subspace Distance (Stewart and Sun, 1990; Collins et al., 2021))** *Let* $\Phi, \Phi_\star \in \mathbb{R}^{r \times d_x}$ *be matrices whose rows are orthonormal. Furthermore, let* $\Phi_{\star, \perp} \in \mathbb{R}^{(d_x - r) \times d_x}$ *be a matrix such that* $\begin{bmatrix} \Phi_\star^\top & \Phi_{\star, \perp}^\top \end{bmatrix}$ *is an orthogonal matrix. Define the distance between the subspaces spanned by the rows of* $\Phi$ *and* $\Phi_\star$ *by*

$$\text{dist}(\Phi, \Phi_\star) := \left\| \Phi \Phi_{\star, \perp}^\top \right\|_2 \tag{2}$$

In particular, the subspace distance quantitatively captures the alignment between two subspaces, interpolating smoothly between 0 (occurring iff $\text{span}(\Phi_\star) = \text{span}(\hat{\Phi})$) and 1 (occurring iff $\text{span}(\Phi_\star) \perp \text{span}(\hat{\Phi})$). We define the task-specific stacked vector notation by capital letters, e.g.,

$$X^{(t)} = \begin{bmatrix} x_1^{(t)} & \cdots & x_i^{(t)} & \cdots & x_N^{(t)} \end{bmatrix}^\top \in \mathbb{R}^{N \times d_x}.$$

The goal of multi-task operator recovery is to estimate $\{F_\star^{(t)}\}_{t=1}^T$ and $\Phi_\star$ from data collected across multiple tasks $\{(x_i^{(t)}, y_i^{(t)})\}_{i=1}^N$, $t = 1, \ldots, T$. Some prior works (Maurer et al., 2016; Du et al., 2020; Zhang et al., 2023) assume access to an empirical risk minimization oracle, i.e. access to

$$\{\hat{F}^{(t)}\}_{t=1}^T, \hat{\Phi} \in \underset{\{F^{(t)}\}, \Phi}{\text{argmin}} \sum_{t=1}^T \sum_{i=1}^N \left\| y_i^{(t)} - F^{(t)} \Phi x_i^{(t)} \right\|^2,$$

focusing on the statistical generalization properties of an ERM solution. However, the above optimization is non-convex even in the linear setting, and thus it is imperative to design and analyze efficient algorithms for recovering optimal matrices $\{F_\star^{(t)}\}_{t=1}^T$ and $\Phi_\star$. To address this problem in the linear regression setting, various works, e.g. `FedRep` (Collins et al., 2021), `AltGD-Min` (Nayer and Vaswani, 2022), propose an alternating minimization-descent scheme, where on a fresh data batch, the weights $\{\hat{F}^{(t)}\}$ are computed on local data via least-squares, and subsequently an estimate of the representation gradient is then computed with respect to local data and aggregated across tasks to perform gradient descent. This algorithmic framework is intuitive, and thus forms a reasonable starting point toward a provably sample-efficient algorithm in our setting.

## 3 SAMPLE-EFFICIENT LINEAR REPRESENTATION LEARNING

We begin by describing the vanilla alternating minimization-descent scheme proposed in Collins et al. (2021). We show that in our setting with label noise and non-isotropy, interaction terms arise in the representation gradient, which cause biases to form that do not scale down with the number of tasks $T$. In §3.2, we propose alterations to the scheme to remove these biases, which we then show in §3.3 lead to fast convergence rates that allow us to recover near-oracle ERM generalization bounds.

### 3.1 PERILS OF (VANILLA) GRADIENT DESCENT ON THE REPRESENTATION

We begin with a summary of the main components of an alternating minimization-descent method analogous to `FedRep` (Collins et al., 2021) and `AltGD-Min` (Nayer and Vaswani, 2022). During each optimization round, a new data batch is sampled for each task: $\{(x_i^{(t)}, y_i^{(t)})\}_{i=1}^N$, $t \in [T]$. We then compute task-specific weights $\hat{F}^{(t)}$ on the corresponding dataset, keeping the current representation estimate $\hat{\Phi}$ fixed. For example, $\hat{F}^{(t)}$ may be the least-squares weights conditioned on $\hat{\Phi}$ (Collins et al., 2021). Define $z_i^{(t)} := \hat{\Phi} x_i^{(t)}$, and the empirical covariance matrices $\hat{\Sigma}_x^{(t)} := \frac{1}{N} X^{(t)\top} X^{(t)}$, $\hat{\Sigma}_z^{(t)} := \frac{1}{N} Z^{(t)\top} Z^{(t)}$. The least squares solution $\hat{F}^{(t)}$ is given by the convex quadratic minimization $\hat{F}^{(t)} = \text{argmin}_F \sum_{i=1}^N \|y_i^{(t)} - F z_i^{(t)}\|^2$. For each task, we then fix the weight matrix $\hat{F}^{(t)}$ and perform a descent step with respect to the representation conditioned on the local data. The resulting representations are averaged across tasks to form the new representation. When the descent direction is the gradient, the update rule is given by

$$\overline{\Phi}_+^{(t)} = \hat{\Phi} - \frac{\eta}{2N} \nabla_\Phi \left\| \sum_{i=1}^N y_i^{(t)} - \hat{F}^{(t)} \hat{\Phi} x_i^{(t)} \right\|^2, \quad \overline{\Phi}_+ = \frac{1}{T} \sum_{t=1}^T \overline{\Phi}_+^{(t)} \tag{3}$$

where $\eta > 0$ is a given step size. We normalize $\overline{\Phi}_+$ to have orthonormal rows, e.g. by (thin/reduced) QR decomposition (Trefethen and Bau, 2022), to produce the final output $\hat{\Phi}_+$, i.e. $\overline{\Phi}_+ = R\hat{\Phi}_+$, $R \in \mathbb{R}^{r \times r}$, leading to

$$R\hat{\Phi}_+ = \hat{\Phi} - \frac{\eta}{T}\sum_{t=1}^{T}\hat{F}^{(t)\top}\left(\hat{F}^{(t)}\hat{\Phi} - F_\star^{(t)}\Phi_\star\right)\hat{\Sigma}_x^{(t)} - \frac{\eta}{NT}\sum_{t=1}^{T}\hat{F}^{(t)\top}W^{(t)\top}X^{(t)}. \tag{4}$$

As in Collins et al. (2021), we right-multiply both sides of (4) by $\Phi_{\star,\perp}^\top$, recalling $\|\Phi\Phi_{\star,\perp}^\top\|_2 =:$ $\text{dist}(\Phi, \Phi_\star)$. Crucially, Collins et al. (2021) assume $x_i^{(t)}$ has mean 0 and identity covariance, and $w_i^{(t)} \equiv 0$ across $i, t$. Therefore, the label noise terms $\hat{F}^{(t)\top}W^{(t)\top}X^{(t)}$ disappear, and the sample covariance for each task $\hat{\Sigma}_x^{(t)}$ concentrates to identity, such that $\Phi_\star\hat{\Sigma}_x^{(t)}\Phi_{\star,\perp}^\top \approx 0$. Under appropriate choice of $\eta$ and bounding the effect of the orthonormalization factor $R$, linear convergence to the optimal representation can be established. However, two issues arise when label noise $w_i^{(t)}$ is introduced and when $x_i^{(t)}$ has non-identity covariance.

1. When *label noise* $w_i^{(t)}$ is present, since $\hat{F}^{(t)}$ is computed on $Y^{(t)}, X^{(t)}$, the gradient noise term is generally biased: $\frac{1}{NT}\mathbb{E}[\hat{F}^{(t)}W^{(t)\top}X^{(t)}] \neq 0$. Even in the simple case that all task distributions $\mathbb{P}_{x,y}^{(t)}$ are identical, $\frac{\eta}{NT}\sum_{t=1}^{T}\hat{F}^{(t)\top}W^{(t)\top}X^{(t)}$ concentrates to its bias, and thus for large $T$ the size of noise term is bottlenecked at $\frac{\eta}{NT}\mathbb{E}\left[\left\|\hat{F}^{(t)\top}W^{(t)\top}X^{(t)}\right\|\right]$. This critically causes the noise term to lose scaling in the *number of tasks* $T$, even when the tasks are identical.
2. When $x_i^{(t)}$ has *non-identity covariance*, the decomposition into a contraction and covariance concentration term no longer holds, since generally $\Phi_*\mathbb{E}[\hat{\Sigma}_x^{(t)}]\Phi_{\star,\perp}^\top \neq 0$. This causes a term whose norm otherwise concentrates around 0 in the isotropic case to scale with $\lambda_{\max}(\hat{\Sigma}_x^{(t)}) - \lambda_{\min}(\hat{\Sigma}_x^{(t)})$ in the worst case.

This motivates modifying the representation update beyond following the vanilla stochastic gradient.

### 3.2 A TASK-EFFICIENT ALGORITHM: DE−BIAS & FEATURE−WHITEN

In the previous section, we identified two fundamental issues: 1. the bias introduced by computing the least squares weights and representation update on the same data batch, and 2. the nuisance term introduced by non-identity second moments of the covariates $x_i^{(t)}$. Toward addressing the first issue, we introduce a "de-biasing" step, where each agent computes the least squares weights $\hat{F}^{(t)}$ and the representation update on *independent* batches of data, e.g. disjoint subsets of trajectories. To address the second issue, we introduce a "feature-whitening" adaptation (LeCun et al., 2002), where the gradient estimate sent by each agent is pre-conditioned by its inverse sample covariance matrix. Combining these two adaptations, the representation update becomes

$$R\hat{\Phi}_+ = \hat{\Phi} - \frac{\eta}{T}\sum_{t=1}^{T}\hat{F}^{(t)\top}\left(\hat{F}^{(t)}\hat{\Phi} - F_\star^{(t)}\Phi_\star\right) - \frac{\eta}{T}\sum_{t=1}^{T}\hat{F}^{(t)\top}W^{(t)\top}X^{(t)}\left(\hat{\Sigma}_x^{(t)}\right)^{-1}, \tag{5}$$

where we assume $\{\hat{F}^{(t)}\}$ are computed on independent data using the aforementioned batching strategy. When $x_i^{(t)}, w_i^{(t)}, t = 1, \dots, T$, are all mutually independent, then the first two terms of the update form the contraction, and the last term is an average of *zero-mean* least-squares-error-like terms over tasks, which can be studied using standard tools (Abbasi-Yadkori et al., 2011; Abbasi-Yadkori, 2013). This culminates in convergence rates that scale favorably with the number of tasks (§3.3). To operationalize our proposed adaptations, let $D^{(t)} = \{(x_i^{(t)}, y_i^{(t)})\}_{i=1}^{N}, t \in [T]$, be a dataset available to each agent. For the weights de-biasing step, we sub-sample trajectories $\mathcal{N}_1 \subset [N]$. For each agent, we compute least-squares weights from $\mathcal{N}_1$. We then sub-sample trajectories $\mathcal{N}_2 \subset [N] \setminus \mathcal{N}_1$, and compute the task-conditioned representation gradients from $\mathcal{N}_2$.

$$\hat{F}^{(t)} = \operatorname*{argmin}_{F}\sum_{i \in \mathcal{N}_1}\left\|y_i^{(t)} - Fz_i^{(t)}\right\|^2, \quad \hat{\mathcal{G}}_{\mathcal{N}_2}^{(t)} = \nabla_\Phi \frac{1}{2}\left\|\sum_{i \in \mathcal{N}_2}y_i^{(t)} - \hat{F}^{(t)}\hat{\Phi}x_i^{(t)}\right\|^2. \tag{6}$$

---

**Algorithm 1** `De-biased & Feature-whitened` (DFW) Alt. Minimization-Descent

---

1: **Input:** step sizes $\{\eta_k\}_{k\geq 1}$, batch sizes $\{N_k\}_{k\geq 1}$, initial estimate $\hat{\Phi}_0$.
2: **for** $k = 1, \ldots, K$ **do**
3:      **for** $t \in [T]$ **(in parallel) do**
4:          Obtain samples $\{(x_i^{(t)}, y_i^{(t)})\}_{i=1}^{N_k}$.
5:          Partition trajectories $[N_k] = \mathcal{N}_{k,1} \sqcup \mathcal{N}_{k,2}$.
6:          Compute $\hat{F}_k^{(t)}$, e.g. via least squares on $\mathcal{N}_{k,1}$ (6).
7:          Compute task-conditioned representation gradient $\hat{\mathcal{G}}_{\mathcal{N}_{k,2}}^{(t)}$ on $\mathcal{N}_{k,2}$ (6).
8:          Compute task-conditioned representation update $\bar{\Phi}_k^{(t)}$ (7).
9:      **end for**
10:     $\hat{\Phi}_k, \_\_ \leftarrow \texttt{thin\_QR}\left(\frac{1}{T}\sum_{t=1}^{T}\bar{\Phi}_k^{(t)}\right)$.
11: **end for**
12: **return** Representation estimate $\hat{\Phi}_K$.

---

Lastly, each agent updates its local representation via a feature-whitened gradient step to yield $\bar{\Phi}_+^{(t)}$. The global representation update is computed by averaging the updated task-conditioned representations $\bar{\Phi}_+^{(t)}$ and performing orthonormalization:

$$\bar{\Phi}_+^{(t)} := \hat{\Phi} - \eta\hat{\mathcal{G}}_{\mathcal{N}_2}^{(t)}\left(\hat{\Sigma}_{x,\mathcal{N}_2}^{(t)}\right)^{-1}, \quad R\hat{\Phi}_+ = \frac{1}{T}\sum_{t=1}^{T}\bar{\Phi}_+^{(t)}, \tag{7}$$

We summarize the full algorithm in Algorithm 1. The above de-biasing and feature whitening steps ensure that the expectation of the representation update is a contraction (with high probability):

$$\mathbb{E}\left[\mathrm{dist}(\hat{\Phi}_+, \Phi_\star)\right] = \mathbb{E}\left[\left\|R^{-1}\left(I_{d_x} - \frac{\eta}{T}\sum_{t=1}^{T}\hat{F}^{(t)\top}\hat{F}^{(t)}\right)\right\|\right]\mathrm{dist}(\hat{\Phi}, \Phi_\star), \tag{8}$$

where the task and trajectory-wise independence ensures that the variance of the gradient scales inversely in $NT$.

### 3.3 ALGORITHM GUARANTEES

We present our main result in the form of convergence guarantees for Algorithm 1. We begin by defining a standard measure of dependency along covariate sequences via $\beta$-mixing.

**Definition 3.1 ($\beta$-mixing)** *Let $\{x_i\}_{i\geq 1}$ be a $\mathbb{R}^d$-valued discrete-time stochastic process adapted to filtration $\{\mathcal{F}_i\}_{i=1}^{\infty}$. We denote the stationary distribution $\nu_\infty$. We define the $\beta$-mixing coefficient*

$$\beta(k) := \sup_{i\geq 1}\mathbb{E}_{\{x_\ell\}_{\ell=1}^{i}}\left[\left\|\mathbb{P}_{x_{i+k}}(\cdot \mid \mathcal{F}_i) - \nu_\infty\right\|_{\mathrm{tv}}\right], \tag{9}$$

*where $\|\cdot\|_{\mathrm{tv}}$ denotes the total variation distance between probability measures.*

Intuitively, the $\beta$-mixing coefficient measures how quickly on average a process mixes to its stationary distribution along any sample path. To instantiate our bounds, we make the following assumptions on the covariates.

**Assumption 3.1 (Subgaussian covariates, geometric mixing)** *We assume the marginal distributions of $x_i^{(t)}$ to be identical with zero-mean, covariance $\Sigma_x^{(t)}$, and to $\gamma^2$-subgaussian across all $i, t$:*

$$\mathbb{E}[x_i^{(t)}] = 0, \quad \mathbb{E}\left[\exp\left(\lambda v^\top x_i^{(t)}\right)\right] \leq \exp\left(\lambda^2 \|v\|^2 \gamma^2\right) \quad a.s. \ \forall \lambda \in \mathbb{R}, v \in \mathbb{R}^{d_x}.$$

*Furthermore, we assume the process $\{x_i^{(t)}\}_{i\geq 1}$ is geometrically $\beta$-mixing with $\beta^{(t)}(k) \leq \Gamma^{(t)}\mu^{(t)k}$, $\Gamma^{(t)} > 0$, $\mu^{(t)} \in [0, 1)$, for each task $t \in [T]$. Lastly, we define $\tau_{\mathrm{mix}}^{(t)} := \left(\frac{\log(\Gamma^{(t)}N/\delta)}{\log(1/\mu^{(t)})} \vee 1\right)$.*

Notably, these assumptions subsume fundamental problems in learning over (stable) dynamical systems, in particular linear system identification and imitation learning, where non-iid and non-isotropy across tasks are unavoidable. We discuss these applications in depth in Appendix B. As in prior work, our final convergence rates depend on a notion of task diversity.

**Definition 3.2 (Task diversity)** *We define the quantities*

$$\lambda_{\min}^{\mathbf{F}} := \lambda_{\min}\left(\frac{1}{T}\sum_{t=1}^{T}F_{\star}^{(t)\top}F_{\star}^{(t)}\right), \quad \lambda_{\max}^{\mathbf{F}} := \lambda_{\max}\left(\frac{1}{T}\sum_{t=1}^{T}F_{\star}^{(t)\top}F_{\star}^{(t)}\right). \quad (10)$$

We now present our main result.

**Theorem 3.1 (Main result)** *Let Assumption 3.1 hold. Assume the current representation iterate $\hat{\Phi}$ satisfies $\mathrm{dist}(\hat{\Phi}, \Phi_{\star}) < \frac{1}{30}\frac{\lambda_{\min}^{\mathbf{F}}}{\sqrt{\lambda_{\max}^{\mathbf{F}}}}\min_t \kappa(\Sigma_x^{(t)})^{-1}$, and*

$$|\mathcal{N}_1| \geq \max_t \tau_{\mathsf{mix}}^{(t)}\,\tilde{\Omega}\left(\max\left\{\gamma^4\left(r + \log(T/\delta)\right), \frac{\lambda_{\max}^{\mathbf{F}}}{(\lambda_{\min}^{\mathbf{F}})^2}\frac{\sigma_w^{(t)2}}{\|F_{\star}^{(t)}\|^2\lambda_{\min}(\Sigma_x^{(t)})}\left(d_y + r + \log(T/\delta)\right)\right\}\right)$$

$$|\mathcal{N}_2|T \geq \max_t \tau_{\mathsf{mix}}^{(t)}\,\tilde{\Omega}\left(\max\left\{T\gamma^4\left(d_x + \log(T/\delta)\right), \frac{\lambda_{\max}^{\mathbf{F}}}{(\lambda_{\min}^{\mathbf{F}})^2}\frac{\sigma_w^{(t)2}}{\|F_{\star}^{(t)}\|^2\lambda_{\min}(\Sigma_x^{(t)})}\left(d_x + r + \log(T/\delta)\right)\right\}\right).$$

*Given $\eta \leq \frac{1}{3\lambda_{\max}^{\mathbf{F}}}$, with probability greater than $1 - \delta$, the next iterate outputted by DFW (Algorithm 1) satisfies*

$$\mathrm{dist}\left(\hat{\Phi}_+, \Phi_{\star}\right) \leq \left(1 - \frac{\eta}{2}\lambda_{\min}^{\mathbf{F}}\right)^{1/2}\mathrm{dist}\left(\hat{\Phi}, \Phi_{\star}\right) + \tilde{\mathcal{O}}\left(\sigma_{\mathrm{avg}}\sqrt{\frac{d_x + \log\left(\frac{T}{\delta}\right)}{NT}}\right),$$

*where $\sigma_{\mathrm{avg}} := \sqrt{\frac{1}{T}\sum_{t=1}^{T}\frac{\sigma_w^{(t)2}\|F_{\star}^{(t)}\|^2}{\lambda_{\min}(\Sigma_x^{(t)})}}$ is the* task-averaged *noise-level.*

**Remark 3.1 (Initialization)** *We note that Theorem 3.1 relies on the representation $\hat{\Phi}$ being sufficiently close to $\Phi_{\star}$. We do not address this issue in this paper, and refer to Collins et al. (2021); Tripuraneni et al. (2021); Thekumparampil et al. (2021) for initialization schemes in the iid linear regression setting. Our experiments suggest that initialization is often unnecessary, which mirrors the experimental findings in Thekumparampil et al. (2021, Sec. 6). We leave constructing an initialization scheme for our general setting, or showing it is unnecessary, to future work.*

A key benefit of having the variance term in Theorem 3.1 scale properly in $N, T$ is that we may construct representations on fixed datasets whose error scales on the same order as the oracle empirical risk minimizer by running Algorithm 1 on appropriately partitioned subsets of a given dataset.

**Corollary 3.1 (Approximate ERM)** *Let the assumptions of Theorem 3.1 hold. Let $\hat{\Phi}_0$ be an initial representation satisfying $\mathrm{dist}(\hat{\Phi}_0, \Phi_{\star}) < \nu$, and define $\rho := \frac{3}{64}\frac{\lambda_{\min}^{\mathbf{F}}}{\lambda_{\max}^{\mathbf{F}}}$. Let $\mathbf{D} := \{\{(x_i^{(t)}, y_i^{(t)})\}_{i=1}^{N}\}_{t\in[T]}$ be a given dataset. There exists a partition of $\mathbf{D}$ into independent batches $\mathbf{B}_1, \ldots, \mathbf{B}_K$, such that iterating DFW on $\mathbf{B}_k$, $k \in [K]$ yields with probability greater than $1 - \delta$:*

$$\mathrm{dist}(\hat{\Phi}_K, \Phi_{\star})^2 \leq \tilde{\mathcal{O}}\left(C(\rho)\frac{\max_h \sigma_w^{(t)2}(d_x r + \log(1/\delta))}{NT}\right), \quad (11)$$

*where $C(\rho) > 0$ is a constant depending on the contraction rate $\rho$.*

We note that the RHS of (11) has the "correct" scaling: noise level $\sigma_w^{(t)2}$ multiplied by # parameters of the representation, divided by the *total* amount of data $NT$. In particular, given a fine-tuning dataset of size $N'$ sampled from a task $T + 1$ that shares the representation $\Phi_{\star}$, computing the least

squares weights $\hat{F}^{(T+1)}$ conditioned on $\hat{\Phi}_K$ yields a high-probability bound (cf. Appendix A.4.1) on the parameter error

$$
\begin{aligned}
\left\| \hat{F}^{(T+1)}\hat{\Phi}_K - M_\star^{(T+1)} \right\|_F^2 &\lesssim \mathrm{dist}(\hat{\Phi}_K, \Phi_\star)^2 + \sigma_w^{(T+1)2}\frac{d_y r + \log(1/\delta)}{N'} \\
&\lesssim C(\rho)\frac{\max_t \sigma_w^{(t)2}(d_x r + \log(1/\delta))}{NT} + \frac{\sigma_w^{(T+1)2}(d_y r + \log(1/\delta))}{N'},
\end{aligned}
$$

where we omit task-related quantities for clarity. We note that the above parameter recovery bound mirrors that of ERM estimates (Du et al., 2020; Zhang et al., 2023), where we note the latter fine-tuning term scales with $d_y r$ (the number of parameters in $F^{(T+1)}$) as opposed to $r$ in the linear regression setting ($d_y = 1$).

## 4 NUMERICAL VALIDATION

We present numerical experiments to demonstrate the key importance of aspects of our proposed algorithm. We consider two scenarios: 1. linear regression with non-isotropic *iid* data, and 2. linear system identification. The linear regression experiments highlight the breakdown of standard approaches when approached with mildly non-isotropic data, thus highlighting the necessity of our proposed `De-biasing & Feature-whitening` steps, and our system identification experiments demonstrate the applicability of our algorithm to sequentially dependent (non-isotropic) data. Additional experiments and experiment details can be found in Appendix C.

### 4.1 LINEAR REGRESSION WITH IID AND NON-ISOTROPIC DATA

We consider the observation model from (1), where we set the operator dimensions and rank as $d_x = d_y = 50$ and $r = 7$. We generate the $T$ operators using the following steps: 1. a ground truth representation $\Phi_\star \in \mathbb{R}^{r \times d_x}$ is randomly generated through applying `thin_svd` to a random matrix with values $\overset{\text{i.i.d.}}{\sim} \mathcal{N}(0,1)$, 2. a nominal task weight matrix $F_0 \in \mathbb{R}^{d_y \times r}$ is generated with elements $\overset{\text{i.i.d.}}{\sim} \mathcal{N}(0,1)$, 3. task-specific weights $F_\star^{(t)} \in \mathbb{R}^{d_y \times r}$, $t \in [T]$ are generated by applying random rotations to $F_0$. A non-isotropic covariance matrix $\Sigma_x$ shared across all tasks is generated as $\Sigma_x = \frac{d_x \tilde{\Sigma}_x}{\mathrm{Tr}(\tilde{\Sigma}_x)}$, where $\tilde{\Sigma}_x = \frac{1}{2}(U + U^\top)$, $U = 5 \cdot I_{d_x} + V$, with $V_{i,j} \overset{\text{i.i.d.}}{\sim} \mathrm{Unif}(0,1)$. We note that by design of $U$, $\Sigma_x$ is only mildly non-isotropic. Figure 1a compares the performance of Algorithm 1, for a single-task ($T = 1$) and multiple tasks ($T = 25$), as well as standard alternating minimization-descent like `FedRep` (Collins et al., 2021) with $T = 25$. For each optimization iteration, we sample $N = 100$ fresh data per task. The figure highlights that `DFW` is able to make use of all source data to decrease variance and learn a representation to near-optimality. As predicted in §3.1, Figure 1a also shows that vanilla alternating minimization-descent is not able to improve beyond a highly suboptimal bias, despite all tasks sharing the same, rather mildly non-isotropic covariate distribution.

For a second experiment, we consider instead of applying vanilla alternating minimization-descent on a linear representation, we parameterize the representation by a neural network. In particular, we consider a ReLU-activated network with one hidden layer of dimension 64. We keep the same data-generation, with $N = 100$ fresh samples per optimization iteration per task, and compare `DFW` (on a linear representation) to AMD on the neural net representation. Since the subspace distance is no longer defined for the neural net representation, we measure optimality of the learned representations by computing the average least-squares loss with respect to a validation set generated through the nominal weight matrix $F_0$. We plot the "optimal" loss as that attained by the ground truth operator $F_0\Phi_\star$. We see in Figure 1b that, despite the much greater feature representation power of the neural net, alternating minimization-descent plateaus for many iterations just like the linear case before finding a non-linear parameter representation that allows descent to optimality, taking almost two orders of magnitude more iterations than `DFW` to reach optimality. This feature learning phase is only exacerbated when there are fewer tasks. We note that in such a streaming data setting, the high iteration complexity of AMD translates to a greatly hampered sample-efficiency compared to `DFW`.

### 4.2 LINEAR SYSTEM IDENTIFICATION

We consider a discrete-time linear system identification (sysID) problem, with dynamics

$$
x_{i+1} = Ax_i + Bu_i + w_i, \ \ i = 0, \ldots, N-1,
$$

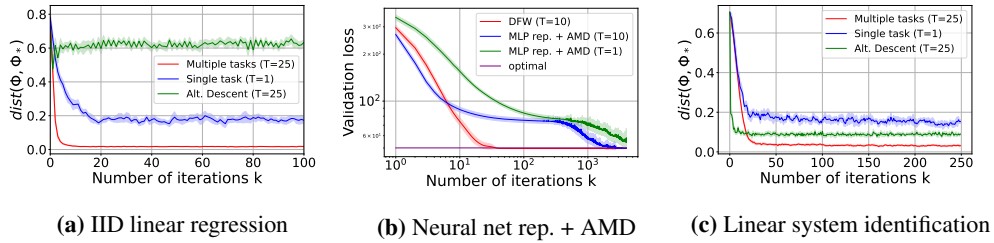

**(a)** IID linear regression     **(b)** Neural net rep. + AMD     **(c)** Linear system identification

**Figure 1:** We plot the suboptimality the current and ground truth representation with respect to the number of iterations, comparing between the single and multiple-task settings of Algorithm 1 and the multi-task alternating minimization-descent. We observe performance improvement and variance reduction for multi-task DFW as predicted. All curves are are plotted as the mean with 95% confidence regions shaded

where $x_i$ is the state of the system and $u_i$ is the control input. In contrast to the previous example, the covariates are now additionally non-iid due to correlation over time. In particular, we can instantiate multi-task linear sysID in the form of (1),

$$x_{i+1}^{(t)} = M_\star^{(t)} z_i^{(t)} + w_i^{(t)}, \;\; i = 0, \ldots, N-1$$

where $M_\star^{(t)} := [A^{(t)} \;\; B^{(t)}] = F^{(t)} \Phi_\star \in \mathbb{R}^{d_x \times d_z}$. The state-action pair at time instant $i$ for all tasks $t \in [T]$ is embedded as $z_i^{(t)} = [x_i^{(t)\top} \;\; u_i^{(t)\top}]^\top$. The process noise $w_i^{(t)}$ and control action $u_i^{(t)}$ are assumed to be drawn from Gaussian distributions $\mathcal{N}(0, \Sigma_w)$ and $\mathcal{N}(0, \sigma_u^2 I_{d_u})$, respectively, where $d_u$ represents the dimension of the control action. We set the state dimension $d_x = 25$, control dimension $d_u = 2$, latent dimension $r = 6$, horizon $N = 100$, and input variance $\sigma_u^2 = 1$. The generation process of the ground truth system matrices $M_\star^{(t)}$ follows a similar approach as described in the linear regression problem, with the addition of a normalization step of the nominal weight matrix $F_0$ to ensure system stability for all tasks $t \in [T]$. Furthermore, the process noise covariance $\Sigma_w$ is parameterized in a similar manner as in the linear regression example, with $U = 5 \cdot I_{d_x} + 2 \cdot V$. The initial state $x_0^{(t)}$ is drawn iid across tasks from the system's stationary distribution $\mathcal{N}(0, \Sigma_x^{(t)})$, which is determined by the solution to the discrete Lyapunov equation $\Sigma_x^{(t)} = A^{(t)} \Sigma_x^{(t)} (A^{(t)})^\top + \sigma_u^2 B^{(t)} (B^{(t)})^\top + \Sigma_w$. We note this implies the covariates $x_i^{(t)}$ are inherently non-isotropic and non-identically distributed across tasks. Figure 1c again demonstrates the advantage of leveraging multi-task data to reduce the error in computing a shared representation across the system matrices $M_\star^{(t)}$. In line with our theoretical findings, DFW continues to benefit from multiple tasks, even when the data is sequentially dependent. We see that FedRep remains suboptimal in this non-iid, non-isotropic setting.

## 5 DISCUSSION AND FUTURE WORK

We propose an efficient algorithm to provably recover linear operators across multiple tasks to optimality from non-iid non-isotropic data, recovering near oracle empirical risk minimization rates. We show that the benefit of learning over multiple tasks manifests in a lower noise level in the optimization and smaller sample requirements for individual tasks. These results contribute toward a general understanding of representation learning from an algorithmic and statistical perspective. Some immediate open questions are: whether good initialization of the representation is necessary, and whether the convergence rate of DFW can be optimized e.g., through $\ell^2$-regularized weights $\hat{F}^{(t)}$. Resolving these questions has important implications for the natural extension of our framework: as emphasized in (Collins et al., 2021), the alternating empirical risk minimization (holding representation fixed) and gradient descent (holding task-specific weights fixed) framework naturally extends to the nonlinear setting. Providing guarantees for nonlinear function classes is an exciting and impactful avenue for future work, which concurrent work is moving toward, e.g. for 2-layer ReLU networks (Collins et al., 2023) and kernel ridge regression (Meunier et al., 2023). It remains to be seen whether a computationally-efficient algorithm can be established for nonlinear meta-learning in the non-iid and non-isotropic data regime, while preserving joint scaling in number of tasks and data.

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
