CONTENTS

# A  THEORETICAL ANALYSIS OF DFW (ALGORITHM 1)

## A.1  PRELIMINARIES

We introduce some preliminary concepts and results that recur throughout our analysis. A fundamental concept in the analysis of least-squares solutions is the self-normalized martingale (Abbasi-Yadkori et al., 2011; Abbasi-Yadkori, 2013).

**Lemma A.1 (cf. Zhang et al. (2023, Lemma B.3))** *Let $\{x_i\}_{i\geq 1}$ be a $\mathbb{R}^{d_x}$-valued process adapted to a filtration $\{\mathcal{F}_i\}_{i\geq 1}$. Let $\{w_i\}_{i\geq 1}$ be a $\mathbb{R}^{d_y}$-valued process adapted to $\{\mathcal{F}_i\}_{i\geq 2}$. Suppose that $\{w_i\}_{i\geq 1}$ is a $\sigma^2$-subgaussian martingale difference sequence, i.e.,:*

$$\mathbb{E}[w_i \mid \mathcal{F}_i] = 0, \tag{12}$$

$$\mathbb{E}[\exp(\lambda v^\top w_i) \mid \mathcal{F}_i] \leq \exp\left(\frac{\lambda^2 \sigma^2 \|v\|^2}{2}\right) \quad \forall \mathcal{F}_i\text{-measurable } \lambda \in \mathbb{R}, v \in \mathbb{R}^{d_y}. \tag{13}$$

*For $\Lambda \in \mathbb{R}^{d_y \times d_x}$, let $\{M_k(\Lambda)\}_{k\geq 1}$ be the $\mathbb{R}$-valued process:*

$$M_k(\Lambda) = \exp\left(\frac{1}{\sigma}\sum_{i=1}^{k}\langle \Lambda x_i, w_i\rangle - \frac{1}{2}\sum_{i=1}^{k}\|\Lambda x_i\|^2\right). \tag{14}$$

*Then, the process $\{M_k(\Lambda)\}_{k\geq 1}$ satisfies $\mathbb{E}[M_k(\Lambda)] \leq 1$ for all $k \geq 1$.*

In particular, this implies the following self-normalized martingale inequality that handles multiple matrix-valued self-normalized martingales simultaneously. This can be seen as an instantiation of the Hilbert space variant from Abbasi-Yadkori (2013).

**Proposition A.1 (cf. Zhang et al. (2023, Prop. B.1), Sarkar et al. (2021, Proposition 8.2))** *Fix $T \in \mathbb{N}_+$. For $t \in [T]$, let $\{x_i^{(t)}, w_i^{(t)}\}_{i\geq 1}$ be a $\mathbb{R}^{d_x} \times \mathbb{R}^{d_y}$-valued process and $\{\mathcal{F}_i^{(t)}\}_{i\geq 1}$ be a filtration such that $\{x_i^{(t)}\}_{i\geq 1}$ is adapted to $\{\mathcal{F}_i^{(t)}\}_{i\geq 1}$, $\{w_i^{(t)}\}_{i\geq 1}$ is adapted to $\{\mathcal{F}_i^{(t)}\}_{i\geq 2}$, and $\{w_i^{(t)}\}_{i\geq 1}$ is a $\sigma^2$-subgaussian martingale difference sequence. Suppose that for all $t_1 \neq t_2$, the process $\{x_i^{(t_1)}, w_i^{(t_1)}\}$ is independent of $\{x_i^{(t_2)}, w_i^{(t_2)}\}$. Fix (non-random) positive definite matrices $\{V^{(t)}\}_{t=1}^{T}$. For $k \geq 1$ and $t \in [T]$, define $\hat{\Sigma}_k^{(t)} := \sum_{i=1}^{k} x_i x_i^\top$. Then, given any fixed $N, T \in \mathbb{N}_+$, with probability at least $1 - \delta$:*

$$\sum_{t=1}^{T}\left\|\sum_{i=1}^{N} w_i^{(t)} x_i^{(t)\top}\left(V^{(t)} + \hat{\Sigma}_N^{(t)}\right)^{-1/2}\right\|_F^2 \leq d_y\sigma^2\sum_{t=1}^{T}\log\left(\frac{\det\left(V^{(t)} + \hat{\Sigma}_N^{(t)}\right)}{\det(V^{(t)})}\right) + 2\sigma^2\log(1/\delta) \tag{15}$$

*Alternatively, in the spectral norm, we have with probability at least $1 - \delta$,*

$$\sum_{t=1}^{T}\left\|\sum_{i=1}^{N} w_i^{(t)} x_i^{(t)\top}\left(V^{(t)} + \hat{\Sigma}_N^{(t)}\right)^{-1/2}\right\|_2^2 \leq 4\sigma^2\sum_{t=1}^{T}\log\left(\frac{\det\left(V^{(t)} + \hat{\Sigma}_N^{(t)}\right)}{\det(V^{(t)})}\right)$$
$$+ 13 d_y T \sigma^2 + 8\sigma^2\log(1/\delta). \tag{16}$$

We note that the above bound also holds for individual tasks $t \in [T]$ simply by removing the summand over $t$. We introduce the following useful two-sided concentration inequality for the sample covariance of iid subgaussian covariates.

**Lemma A.2 (Du et al. (2020, Claim A.1, A.2))** *Let $x_1, \ldots, x_N \in \mathbb{R}^d$ be iid random vectors that satisfy $\mathbb{E}[x_i] = 0$, $\mathbb{E}\left[x_i x_i^\top\right] = \Sigma$, and $x_i$ is $\gamma^2$-subgaussian. Fix $\delta \in (0, 1)$. Suppose $N \gtrsim \gamma^4(d + \log(1/\delta))$. Then with probability at least $1 - \delta$, the following holds*

$$0.9\Sigma \preceq \frac{1}{N}\sum_{i=1}^{N} x_i x_i^\top \preceq 1.1\Sigma. \tag{17}$$

*Furthermore, for any matrix $U \in \mathbb{R}^{r \times d_x}$, as long as $N \gtrsim \gamma^4 \left( r + \log(1/\delta) \right)$, we have with probability at least $1 - \delta$*

$$0.9 U \Sigma U^\top \preceq \frac{1}{N} \sum_{i=1}^{N} U x_i x_i^\top U^\top \preceq 1.1 U \Sigma U^\top. \tag{18}$$

Combining Proposition A.1 and Lemma A.2 yields the following self-normalized martingale bound without the non-random lower bound $V^{(t)}$.

**Lemma A.3** *Consider the quantities defined in Proposition A.1 and assume $x_i^{(t)}$ are zero-mean and $\gamma^2$-subgaussian. Then, as long as $N \gtrsim \gamma^4 \left( d_x + \log(T/\delta) \right)$, with probability at least $1 - \delta$:*

$$\sum_{t=1}^{T} \left\| \sum_{i=1}^{N} w_i^{(t)} x_i^{(t)\top} \left( \hat{\Sigma}_N^{(t)} \right)^{-1/2} \right\|_F^2 \leq 2 d_y d_x T \sigma^2 + 4 \sigma^2 \log(T/\delta), \text{ or}$$

$$\sum_{t=1}^{T} \left\| \sum_{i=1}^{N} w_i^{(t)} x_i^{(t)\top} \left( \hat{\Sigma}_N^{(t)} \right)^{-1/2} \right\|_2^2 \leq 8 d_x T \sigma^2 + 26 d_y T \sigma^2 + 16 \sigma^2 \log(T/\delta)$$

*Proof:* we observe that if $\hat{\Sigma}_N^{(t)} \succeq V^{(t)}$, then

$$2 \hat{\Sigma}_N^{(t)} \succeq V^{(t)} + \hat{\Sigma}_N^{(t)} \implies \left( \hat{\Sigma}_N^{(t)} \right)^{-1} \preceq 2 \left( V^{(t)} + \hat{\Sigma}_N^{(t)} \right)^{-1}.$$

This implies

$$\sum_{t=1}^{T} \mathbf{1} \left\{ \hat{\Sigma}_N^{(t)} \succeq V^{(t)} \right\} \left\| \sum_{i=1}^{N} w_i^{(t)} x_i^{(t)\top} \left( \hat{\Sigma}_N^{(t)} \right)^{-1/2} \right\|^2$$

$$\leq 2 \sum_{t=1}^{T} \mathbf{1} \left\{ \hat{\Sigma}_N^{(t)} \succeq V^{(t)} \right\} \left\| \sum_{i=1}^{N} w_i^{(t)} x_i^{(t)\top} \left( V^{(t)} + \hat{\Sigma}_N^{(t)} \right)^{-1/2} \right\|^2.$$

Defining $\Sigma^{(t)} := \mathbb{E} \left[ x_i^{(t)} x_i^{(t)\top} \right]$, let us consider for each $t$ the event:

$$0.9 \Sigma^{(t)} \preceq \hat{\Sigma}_N^{(t)} \preceq 1.1 \Sigma^{(t)},$$

which by Lemma A.2 occurs with probability at least $1 - \delta$ as long as $N \gtrsim \gamma^4 (d_x + \log(1/\delta))$. Setting $V^{(t)} := 0.9 N \Sigma^{(t)}$ and conditioning on the above event, we observe that by definition $\hat{\Sigma}_N^{(t)} \succeq V^{(t)}$, and

$$\log \frac{\det \left( V^{(t)} + \hat{\Sigma}_N^{(t)} \right)}{\det(V^{(t)})} = \log \det \left( I_{d_x} + \hat{\Sigma}_N^{(t)} \left( V^{(t)} \right)^{-1} \right)$$

$$= \log \det \left( \left( 1 + \frac{1.1}{0.9} \right) I_{d_x} \right)$$

$$\leq d_x.$$

Plugging this into Proposition A.1 and union-bounding over the desirable event over each $t \in [T]$, and adjusting the failure probability $\delta/T \mapsto \delta$, we get our desired result. $\blacksquare$

In order to instantiate our bounds for non-iid covariates, we introduce the notions of $\beta$-mixing stationary processes (Kuznetsov and Mohri, 2017; Tu and Recht, 2018).

**Definition A.1 ($\beta$-mixing)** *Let $\{x_i\}_{t \geq 1}$ be a $\mathbb{R}^d$-valued discrete-time stochastic process adapted to filtration $\{\mathcal{F}_i\}_{t=1}^{\infty}$. We denote the stationary distribution $\nu_\infty$. We define the $\beta$-mixing coefficient*

$$\beta(k) := \sup_{t \geq 1} \mathbb{E}_{\{x_\ell\}_{\ell=1}^{t}} \left[ \left\| \mathbb{P}_{x_{t+k}}(\cdot \mid \mathcal{F}_i) - \nu_\infty \right\|_{\text{tv}} \right], \tag{19}$$

*where $\|\cdot\|_{\text{tv}}$ denotes the total variation distance between probability measures.*

Intuitively, the $\beta$-mixing coefficient measures how quickly on average a process mixes to the stationary distribution along any sample path. To see how $\beta$-mixing is instantiated, let $\{x_i\}_{t=1}^T$ be a sample path from a $\beta$-mixing process. Consider the following subsampled paths formed by taking every $a$-th covariate of $\{x_i\}$:

$$X_{(j)}^T := \{x_i : 1 \leq t \leq T,\ (t-1 \mod a) = j-1\},\quad j = 1, \ldots, a. \tag{20}$$

Let the integers $m_1, \ldots, m_a$ and index sets $I_{(1)}, \ldots I_{(a)}$ denote the sizes and indices of $X_{(1)}^T, \ldots, X_{(a)}^T$, respectively. Finally, let $X_\infty^{m_j}$ denote a sequence of $m_j$ iid draws from the stationary distribution $\nu_\infty$. The following is a key lemma in relating a correlated process to iid draws.

**Lemma A.4 (Kuznetsov and Mohri (2017, Proposition 2))** *Let $g(\cdot)$ be a real-valued Borel-measurable function satisfying $-M_1 \leq g(\cdot) \leq M_2$ for some $M_1, M_2 \geq 0$. Then, for all $j = 1, \ldots, a$.*

$$\left| \mathbb{E}[g(X_\infty^{m_j})] - \mathbb{E}[g(X_{(j)}^T)] \right| \leq (M_1 + M_2) m_j \beta(a).$$

In our analysis, we often instantiate Lemma A.4 with $g(\cdot)$ as an indicator function on a success event. For appropriately selected block length $a$, we are thus able to relate simpler iid analysis on $X_\infty^{m_j}$ to the original process $X_{(j)}^T$, accruing an additional factor in the failure probability. Lastly, we introduce a standard matrix concentration inequality.

**Lemma A.5 (Matrix Hoeffding (Tropp, 2011))** *Let $\{X_t\}_{t=1}^T \subset \mathbb{R}^{d \times d}$ be a sequence of independent, random symmetric matrices, and let $\{B_t\}_{t=1}^T$ be a sequence of fixed symmetric matrices. Assume each random matrix satisfies*

$$\mathbb{E}[X_t] = 0,\quad X_t^2 \preceq B_t^2 \text{ almost surely.}$$

*Then for all $t \geq 0$,*

$$\mathbb{P}\left[\lambda_{\max}\left(\sum_{t=1}^T X_t\right) \geq t\right] \leq d \cdot \exp\left(-\frac{t^2}{8\sigma^2}\right),\quad \sigma^2 := \left\|\sum_{t=1}^T B_t^2\right\|.$$

*In particular, for general rectangular $\{M_t\}_{t=1}^T \subset \mathbb{R}^{d_1 \times d_2}$, we may define $X_t := \begin{bmatrix} 0 & M_t \\ M_t^\top & 0 \end{bmatrix}$ to yield a singular value concentration inequality. Assume each $M_t$ satisfies*

$$\mathbb{E}[M_t] = 0,\quad X_t^2 \preceq B_t^2 \text{ almost surely.}$$

*Then for all $t \geq 0$,*

$$\mathbb{P}\left[\sigma_{\max}\left(\sum_{t=1}^T M_t\right) \geq t\right] \leq (d_1 + d_2) \cdot \exp\left(-\frac{t^2}{8\sigma^2}\right),\quad \sigma^2 := \left\|\sum_{t=1}^T B_t^2\right\|.$$

As hinted by the indexing of the matrices, by leveraging the independence of processes across tasks $h$, Lemma A.5 can be used to bound various quantities averaged across tasks, under the important caveat that the matrices are *zero-mean*, which ties back to the necessity of our de-biasing and feature-whitening adjustments.

## A.2   THE IID SETTING

We recall that given the current representation iterate $\hat{\Phi}$, an iid draw of a multitask dataset $\{(x_i^{(t)}, y_i^{(t)})\}_{t=1, i=1}^{T,N}$, $h = 1, \ldots, H$, and DFW trajectory partitions $\mathcal{N}_1, \mathcal{N}_2$, the least squares weights $\hat{F}^{(t)}$ can be written as

$$
\begin{aligned}
\hat{F}^{(t)} &= \underset{F}{\operatorname{argmin}} \sum_{i \in \mathcal{N}_1} \left\| y_i^{(t)} - F z_i^{(t)} \right\|^2 \\
&= F_\star^{(t)} \Phi_\star X_{\mathcal{N}_1}^{(t)\top} Z_{\mathcal{N}_1}^{(t)} \left(\hat{\Sigma}_{z,\mathcal{N}_1}^{(t)}\right)^{-1} + W_{\mathcal{N}_1}^{(t)\top} Z_{\mathcal{N}_1}^{(t)} \left(\hat{\Sigma}_{z,\mathcal{N}_1}^{(t)}\right)^{-1} \\
&= F_\star^{(t)} \Phi_\star \hat{\Phi}^\top + F_\star^{(t)} \Phi_\star \left(I_{d_x} - \hat{\Phi}^\top \hat{\Phi}\right) X_{\mathcal{N}_1}^{(t)\top} Z_{\mathcal{N}_1}^{(t)} \left(\hat{\Sigma}_{z,\mathcal{N}_1}^{(t)}\right)^{-1} + W_{\mathcal{N}_1}^{(t)\top} Z_{\mathcal{N}_1}^{(t)} \left(\hat{\Sigma}_{z,\mathcal{N}_1}^{(t)}\right)^{-1}.
\end{aligned}
$$
$$\tag{21}$$

Now recalling the `DFW` representation update in the iid setting (5), we have

$$R\hat{\Phi}_+ = \hat{\Phi} - \frac{\eta}{T}\sum_{t=1}^{T}\hat{F}^{(t)\top}\left(\hat{F}^{(t)}\hat{\Phi} - F_\star^{(t)}\Phi_\star\right) - \frac{\eta}{T}\sum_{t=1}^{T}\hat{F}^{(t)\top}W_{\mathcal{N}_2}^{(t)\top}X_{\mathcal{N}_2}^{(t)}\left(\hat{\Sigma}_x^{(t)}\right)^{-1}. \tag{22}$$

Right multiplying the update by $\Phi_{\star,\perp}^\top$, we get

$$R\hat{\Phi}_+\Phi_{\star,\perp}^\top = \hat{\Phi}\Phi_{\star,\perp}^\top - \frac{\eta}{T}\sum_{t=1}^{T}\hat{F}^{(t)\top}\left(\hat{F}^{(t)}\hat{\Phi} - F_\star^{(t)}\Phi_\star\right)\Phi_{\star,\perp}^\top - \frac{\eta}{T}\sum_{t=1}^{T}\hat{F}^{(t)\top}W_{\mathcal{N}_2}^{(t)\top}X_{\mathcal{N}_2}^{(t)}\left(\hat{\Sigma}_{x,\mathcal{N}_2}^{(t)}\right)^{-1}\Phi_{\star,\perp}^\top$$

$$= \left(I_{dx} - \frac{\eta}{T}\sum_{t=1}^{T}\hat{F}^{(t)\top}\hat{F}^{(t)}\right)\hat{\Phi}\Phi_{\star,\perp}^\top - \frac{\eta}{T}\sum_{t=1}^{T}\hat{F}^{(t)\top}W_{\mathcal{N}_2}^{(t)\top}X_{\mathcal{N}_2}^{(t)}\left(\hat{\Sigma}_{x,\mathcal{N}_2}^{(t)}\right)^{-1}\Phi_{\star,\perp}^\top,$$

where the last line is composed of a contraction term and a noise term. We start with an analysis of the noise term.

### BOUNDING THE NOISE TERM

We observe that since $\hat{F}^{(t)}$ is by construction independent of $W_{\mathcal{N}_2}^{(t)}, X_{\mathcal{N}_2}^{(t)}$, by the independence of $x_i^{(t)}$ across $t$ and $i$, and the noise independence $w_i^{(t)} \perp x_i^{(t)}$, we find

$$\mathbb{E}\left[\frac{1}{T}\sum_{t=1}^{T}\hat{F}^{(t)\top}W_{\mathcal{N}_2}^{(t)\top}X_{\mathcal{N}_2}^{(t)}\left(\hat{\Sigma}_{x,\mathcal{N}_2}^{(t)}\right)^{-1}\right] = \frac{1}{T}\sum_{t=1}^{T}\mathbb{E}\left[\hat{F}^{(t)}\right]^\top\mathbb{E}\left[W_{\mathcal{N}_2}^{(t)}\right]\mathbb{E}\left[X_{\mathcal{N}_2}^{(t)}\left(\hat{\Sigma}_{x,\mathcal{N}_2}^{(t)}\right)^{-1}\right] = 0.$$

Therefore, we set up for an application of Lemma A.5. Toward doing so, we prove the following two ingredients: 1. a high probability bound on $\|\hat{F}^{(t)}\|$, 2. a high probability bound on the least-squares noise-esque term $\|\hat{F}^{(t)\top}W_{\mathcal{N}_2}^{(t)\top}X_{\mathcal{N}_2}^{(t)}\left(\hat{\Sigma}_{x,\mathcal{N}_2}^{(t)}\right)^{-1}\|$. We then condition on these two high-probability events to instantiate the almost-sure boundedness in Lemma A.5. We start with the analysis of $\hat{F}^{(t)}$. By (21), we have

$$\left\|\hat{F}^{(t)}\right\| \le \left\|F_\star^{(t)}\right\| + \left\|F_\star^{(t)}\right\|\left\|\Phi_\star\mathcal{P}_{\hat{\Phi}}^\perp X_{\mathcal{N}_1}^{(t)\top}Z_{\mathcal{N}_1}^{(t)}\left(\hat{\Sigma}_{z,\mathcal{N}_1}^{(t)}\right)^{-1}\right\| + \left\|W_{\mathcal{N}_1}^{(t)\top}Z_{\mathcal{N}_1}^{(t)}\left(\hat{\Sigma}_{z,\mathcal{N}_1}^{(t)}\right)^{-1}\right\|.$$

**Lemma A.6** *Let $|\mathcal{N}_1| := N_1 \gtrsim \gamma^4\left(r + \log(1/\delta)\right)$. Then, with probability greater than $1 - \delta$, we have*

$$\left\|\Phi_\star\mathcal{P}_{\hat{\Phi}}^\perp X_{\mathcal{N}_1}^{(t)\top}Z_{\mathcal{N}_1}^{(t)}\left(\hat{\Sigma}_{z,\mathcal{N}_1}^{(t)}\right)^{-1}\right\| \le \frac{5}{4}\mathrm{dist}(\hat{\Phi}, \Phi_\star)\kappa\left(\Sigma_x^{(t)}\right), \tag{23}$$

$$\left\|W_{\mathcal{N}_1}^{(t)\top}Z_{\mathcal{N}_1}^{(t)}\left(\hat{\Sigma}_{z,\mathcal{N}_1}^{(t)}\right)^{-1}\right\| \lesssim \sigma_w^{(t)}\sqrt{\frac{d_y + r + \log(1/\delta)}{\lambda_{\min}(\Sigma_x^{(t)})N_1}}. \tag{24}$$

*Proof:* we begin with the bound (23). We observe that by definition $\frac{1}{N_1}X_{\mathcal{N}_1}^{(t)\top}X_{\mathcal{N}_1}^{(t)} = \hat{\Sigma}_{x,\mathcal{N}_1}^{(t)}$, and $\Phi_\star\mathcal{P}_{\hat{\Phi}}^\perp, \hat{\Phi}$ are $r \times d_x$ matrices. Therefore, we invoke Lemma A.2 twice to find: as long as $N_1 \gtrsim \gamma^4\left(r + \log(1/\delta)\right)$, with probability at least $1 - \delta$, the following bounds hold simultaneously:

$$\frac{1}{N_1}\left\|\Phi_\star\mathcal{P}_{\hat{\Phi}}^\perp X_{\mathcal{N}_1}^{(t)\top}\right\|^2 \le 1.1\left\|\Phi_\star\mathcal{P}_{\hat{\Phi}}^\perp\left(\Sigma_x^{(t)}\right)^{1/2}\right\|^2 \le 1.1\mathrm{dist}(\hat{\Phi}, \Phi_\star)^2\lambda_{\max}\left(\Sigma_x^{(t)}\right)$$

$$\frac{1}{N_1}\left\|Z_{\mathcal{N}_1}^{(t)}\right\|^2 \le 1.1\left\|\Sigma_z^{(t)}\right\| \le 1.1\lambda_{\max}\left(\Sigma_x^{(t)}\right)$$

$$\left\|\left(\hat{\Sigma}_{z,\mathcal{N}_1}^{(t)}\right)^{-1}\right\| \le 0.9\lambda_{\min}(\Sigma_z^{(t)})^{-1} \le 0.9\lambda_{\min}(\Sigma_x^{(t)})^{-1},$$

where we recall that $\Sigma_z^{(t)} = \hat{\Phi}\Sigma_x^{(t)}\hat{\Phi}^\top$. Therefore, applying Cauchy-Schwarz on the LHS of (23) and the above bounds (converting $1.1/0.9 < 5/4$) yields the desired upper bound on the RHS.

Moving onto (24), we observe that since $\hat{\Phi}$ is fixed, $\{w_i^{(t)}, z_i^{(t)}\}_{i \geq 1}$ is an $\mathbb{R}^{d_y} \times \mathbb{R}^r$-valued martingale difference sequence. Therefore, we may apply Lemma A.2 and Lemma A.3 to find: as long as $N_1 \gtrsim \gamma^4 (r + \log(1/\delta))$, with probability at least $1 - \delta$,

$$\left\| W_{\mathcal{N}_1}^{(t)\top} Z_{\mathcal{N}_1}^{(t)} \left( \hat{\Sigma}_{z,\mathcal{N}_1}^{(t)} \right)^{-1} \right\| \leq \left\| W_{\mathcal{N}_1}^{(t)\top} Z_{\mathcal{N}_1}^{(t)} \left( \hat{\Sigma}_{z,\mathcal{N}_1}^{(t)} \right)^{-1/2} \right\| \lambda_{\min} \left( \hat{\Sigma}_{z,\mathcal{N}_1}^{(t)} \right)^{-1/2}$$

$$\lesssim \sqrt{\frac{\sigma_w^{(t)2} (d_y + r + \log(1/\delta))}{\lambda_{\min}(\Sigma_x^{(t)}) N_1}},$$

which establishes the bound (24). ∎

Therefore, by assuming $\mathrm{dist}(\hat{\Phi}, \Phi_\star)$ is sufficiently small, and that $N_1$ is large enough to offset the noise bound (24), we immediately get the following bound relating $\|\hat{F}^{(t)}\|$ to $\|F_\star^{(t)}\|$.

**Lemma A.7** *Assume*

$$N_1 \gtrsim \max \left\{ \gamma^4 (r + \log(1/\delta)), \frac{\sigma_w^{(t)2}}{C^2 \|F_\star^{(t)}\|^2 \lambda_{\min}(\Sigma_x^{(t)})} (d_y + r + \log(1/\delta)) \right\},$$

*and* $\mathrm{dist}(\hat{\Phi}, \Phi_\star)^2 \leq \frac{2}{5} C^2 \kappa \left( \Sigma_x^{(t)} \right)^{-1}$, *for fixed* $C > 0$. *Then with probability at least* $1 - \delta$

$$\|\hat{F}^{(t)}\| \leq (1 + C)\|F_\star^{(t)}\|. \tag{25}$$

The factor $C > 0$ serves as a free parameter which we can determine later–a larger $C$ implies a relaxed requirement on the initial condition (disappears when $C \geq \frac{5}{2} \kappa(\Sigma_x^{(t)})$) and burn-in requirement on $N_1$, but results in a larger subgaussian-parameter bound on the representation update noise term (22), as we demonstrate: given that the success event of Lemma A.7 holds, then by the definition of subgaussianity (Assumption 3.1), we observe that $\hat{F}^{(t)\top} w_i^{(t)}$ is zero-mean and $(1+C)^2 \sigma_w^{(t)2} \|F_\star^{(t)}\|^2$-subgaussian, supported on $\mathbb{R}^r$. Therefore, bounding

$$\left\| \hat{F}^{(t)\top} W_{\mathcal{N}_2}^{(t)\top} X_{\mathcal{N}_2}^{(t)} \left( \hat{\Sigma}_{x,\mathcal{N}_2}^{(t)} \right)^{-1} \right\| \leq \left\| \hat{F}^{(t)} W_{\mathcal{N}_2}^{(t)\top} X_{\mathcal{N}_2}^{(t)} \left( \hat{\Sigma}_{x,\mathcal{N}_2}^{(t)} \right)^{-1/2} \right\| \lambda_{\min}(\hat{\Sigma}_{x,\mathcal{N}_2}^{(t)})^{-1/2},$$

we invoke Proposition A.1 and Lemma A.2 to get the following bound.

**Lemma A.8** *Let the conditions of Lemma A.7 hold. Then, as long as* $|\mathcal{N}_2| := N_2 \gtrsim \gamma^4 (d_x + \log(1/\delta))$, *with probability at least* $1 - \delta$,

$$\left\| \hat{F}^{(t)\top} W_{\mathcal{N}_2}^{(t)\top} X_{\mathcal{N}_2}^{(t)} \left( \hat{\Sigma}_{x,\mathcal{N}_2}^{(t)} \right)^{-1} \right\| \lesssim (1 + C) \sigma_w^{(t)} \|F_\star^{(t)}\| \sqrt{\frac{d_x + \log(1/\delta)}{\lambda_{\min}(\Sigma_x^{(t)}) N_2}}. \tag{26}$$

With a bound on the task-specific noise term in hand, we may now produce the final bound on the (task-averaged) noise term.

**Proposition A.2 (Noise term bound)** *Assume*

$$N_1 \gtrsim \max \left\{ \gamma^4 (r + \log(T/\delta)), \max_t \frac{\sigma_w^{(t)2}}{C^2 \|F_\star^{(t)}\|^2 \lambda_{\min}(\Sigma_x^{(t)})} (d_y + r + \log(T/\delta)) \right\},$$

$$N_2 \gtrsim \gamma^4 (d_x + \log(T/\delta)),$$

*and* $\mathrm{dist}(\hat{\Phi}, \Phi_\star)^2 \leq \max_t \frac{2}{5} C^2 \kappa \left( \Sigma_x^{(t)} \right)^{-1}$, *for fixed* $C > 0$. *Then, with probability at least* $1 - \delta$,

$$\left\| \frac{1}{T} \sum_{t=1}^T \hat{F}^{(t)\top} W_{\mathcal{N}_2}^{(t)\top} X_{\mathcal{N}_2}^{(t)} \left( \hat{\Sigma}_{x,\mathcal{N}_2}^{(t)} \right)^{-1} \right\| \lesssim \sigma_{\mathrm{avg}} (1 + C) \sqrt{\frac{d_x + \log(T/\delta)}{T N_2}} \log \left( \frac{d_y + r}{\delta} \right),$$

*where* $\sigma_{\mathrm{avg}} := \sqrt{\frac{1}{T} \sum_{t=1}^T \frac{\sigma_w^{(t)2} \|F_\star^{(t)}\|^2}{\lambda_{\min}(\Sigma_x^{(t)})}}$ *is the* task-averaged *noise-level.*

*Proof of Proposition A.2:* we set up for an application of the Matrix Hoeffding bound (Lemma A.5). By union bounding over the task-specific noise bound Lemma A.8, we have with probability at least $1 - \delta$, for all $t \in [T]$ simultaneously:

$$\left\| \hat{F}^{(t)\top} W_{\mathcal{N}_2}^{(t)\top} X_{\mathcal{N}_2}^{(t)} \left( \hat{\Sigma}_{x,\mathcal{N}_2}^{(t)} \right)^{-1} \right\| \lesssim (1 + C) \frac{\sigma_w^{(t)} \| F_\star^{(t)} \|}{\sqrt{\lambda_{\min}(\Sigma_x^{(t)})}} \sqrt{\frac{d_x + \log(T/\delta)}{N_2}},$$

given the assumed burn-in conditions hold. Therefore, by setting

$$B^{(t)} = \mathcal{O}(1) \frac{\sigma_w^{(t)} \| F_\star^{(t)} \|}{\sqrt{\lambda_{\min}(\Sigma_x^{(t)})}} \sqrt{\frac{d_x + \log(T/\delta)}{N_2}} I_{d_x + r}$$

$$\sigma^2 = \left\| \sum_{t=1}^{T} (B^{(t)})^2 \right\| = \left( \sum_{t=1}^{T} \frac{\sigma_w^{(t)2} \| F_\star^{(t)} \|^2}{\lambda_{\min}(\Sigma_x^{(t)})} \right) (1 + C)^2 \frac{d_x + \log(T/\delta)}{N_2}$$

$$= T \sigma_{\text{avg}}^2 (1 + C)^2 \frac{d_x + \log(T/\delta)}{N_2}$$

we invoke Lemma A.5 and invert

$$(d_x + r) \cdot \exp\left( \frac{-t^2}{8\sigma^2} \right) \leq \delta$$

to set

$$t \approx \sqrt{T} \sigma_{\text{avg}} (1 + C) \sqrt{\frac{d_x + \log(T/\delta)}{N_2} \log\left( \frac{d_y + r}{\delta} \right)}.$$

The resulting Hoeffding bound yields

$$\mathbb{P}\left[ \left\| \sum_{t=1}^{T} \hat{F}^{(t)\top} W_{\mathcal{N}_2}^{(t)\top} X_{\mathcal{N}_2}^{(t)} \left( \hat{\Sigma}_{x,\mathcal{N}_2}^{(t)} \right)^{-1} \right\| \gtrsim \sqrt{T} \sigma_{\text{avg}} (1 + C) \sqrt{\frac{d_x + \log(T/\delta)}{N_2} \log\left( \frac{d_y + r}{\delta} \right)} \right] \leq \delta$$

$$\iff \mathbb{P}\left[ \left\| \frac{1}{T} \sum_{t=1}^{T} \hat{F}^{(t)\top} W_{\mathcal{N}_2}^{(t)\top} X_{\mathcal{N}_2}^{(t)} \left( \hat{\Sigma}_{x,\mathcal{N}_2}^{(t)} \right)^{-1} \right\| \gtrsim \sigma_{\text{avg}} (1 + C) \sqrt{\frac{d_x + \log(T/\delta)}{T N_2} \log\left( \frac{d_y + r}{\delta} \right)} \right] \leq \delta$$

∎

We have bounded the noise term on the DFW representation update, demonstrating critically the bound on the noise term benefits from a scaling with the number of tasks $T$. The task-relevant quantity $\sigma_{\text{avg}}$ quantifies that the "noise level" of the problem is an *average* over the noise-levels of each task. We note that our application of Matrix Hoeffding is rather crude, and the above bound can likely be improved in terms of $\text{polylog}(1/\delta)$ factors with stronger moment bounds on the matrix-valued self-normalized martingale terms, but this is out of the scope of this paper.

Returning to the choice of $C$, $C \approx 1$ implies no further system/task-specific dependence beyond the terms in $K^{(1:T)}$; however, this may translate into a stringent requirement on the burn-in $N_1$ and the subspace distance $\text{dist}(\hat{\Phi}, \Phi_\star)$. On the other hand, $C \gtrsim \sqrt{\kappa(\Sigma_x^{(t)})}$ relaxes the burn-in and potentially renders the subspace distance requirement trivial, but manifests a condition number in the noise bound. We note that as we expect $\text{dist}(\hat{\Phi}, \Phi_\star)$ to decrease geometrically with iterations of DFW, the subspace distance requirement is only relevant for the first few iterations. In general, this intuitively captures the cost of ill-conditioned data distributions. We now move on to bounding the contraction term.

BOUNDING THE CONTRACTION TERM

Let us define,

$$\Delta^{(t)} := \Phi_\star \left( I_{d_x} - \hat{\Phi}^\top \hat{\Phi} \right) X_{\mathcal{N}_1}^{(t)\top} Z_{\mathcal{N}_1}^{(t)} \left( \hat{\Sigma}_{z,\mathcal{N}_1}^{(t)} \right)^{-1}$$

$$E^{(t)} := W_{\mathcal{N}_1}^{(t)\top} Z_{\mathcal{N}_1}^{(t)} \left( \hat{\Sigma}_{z,\mathcal{N}_1}^{(t)} \right)^{-1},$$

such that we may write (21) as $\hat{F}^{(t)} = F_\star^{(t)} \Phi_\star \hat{\Phi}^\top + F_\star^{(t)} \Delta^{(t)} + E^{(t)}$. We expand

$$\hat{F}^{(t)\top} \hat{F}^{(t)} = \hat{\Phi} \Phi_\star^\top F_\star^{(t)\top} F_\star^{(t)} \Phi_\star \hat{\Phi}^\top + \Delta^{(t)\top} \Delta^{(t)} + E^{(t)\top} E^{(t)}$$
$$+ \operatorname{Sym}(\Delta^{(t)\top} F_\star^{(t)} \Phi_\star \hat{\Phi}^\top) + \operatorname{Sym}(E^{(t)\top} F_\star^{(t)} \Phi_\star \hat{\Phi}^\top) + \operatorname{Sym}(\Delta^{(t)\top} E^{(t)}),$$

where $\operatorname{Sym}(A) := A + A^\top$. We will make repeated use of the following matrix Cauchy-Schwarz-type lemma.

**Lemma A.9** *Let $A_t, B_t$ be real-valued matrices for $t = 1, \dots, T$. Then,*

$$\left\| \sum_{t=1}^T A_t B_t \right\| \le \left\| \sum_{t=1}^T A_t A_t^\top \right\|^{1/2} \left\| \sum_{t=1}^T B_t^\top B_t \right\|^{1/2}.$$

Now taking the average over tasks $T$, we then may write

$$\lambda_{\max} \left( \frac{1}{T} \sum_{t=1}^T \hat{F}^{(t)\top} \hat{F}^{(t)} \right)$$

$$\le \lambda_{\max} \left( \frac{1}{T} \sum_{t=1}^T F_\star^{(t)\top} F_\star^{(t)} \right) + \lambda_{\max} \left( \frac{1}{T} \sum_{t=1}^T \Delta^{(t)\top} \Delta^{(t)} \right) + \lambda_{\max} \left( \frac{1}{T} \sum_{t=1}^T E^{(t)\top} E^{(t)} \right)$$

$$+ \left\| \operatorname{Sym}(\frac{1}{T} \sum_{t=1}^T \Delta^{(t)\top} F_\star^{(t)} \Phi_\star \hat{\Phi}^\top) \right\| + \left\| \operatorname{Sym}(\frac{1}{T} \sum_{t=1}^T E^{(t)\top} F_\star^{(t)} \Phi_\star \hat{\Phi}^\top) \right\| + \left\| \operatorname{Sym}(\frac{1}{T} \sum_{t=1}^T \Delta^{(t)\top} E^{(t)}) \right\|.$$

We observe that

$$\|\operatorname{Sym}(A)\| = \max_{\|u\|, \|v\|=1} x^\top A v + x^\top A^\top v$$
$$\le 2 \|A\|,$$

and thus applying the above fact and Lemma A.9 on the cross terms yields

$$\lambda_{\max} \left( \frac{1}{T} \sum_{t=1}^T \hat{F}^{(t)\top} \hat{F}^{(t)} \right)$$

$$\le \lambda_{\max} \left( \frac{1}{T} \sum_{t=1}^T F_\star^{(t)\top} F_\star^{(t)} \right) + \lambda_{\max} \left( \frac{1}{T} \sum_{t=1}^T \Delta^{(t)\top} \Delta^{(t)} \right) + \lambda_{\max} \left( \frac{1}{T} \sum_{t=1}^T E^{(t)\top} E^{(t)} \right)$$

$$+ 2 \left\| \frac{1}{T} \sum_{t=1}^T \Delta^{(t)\top} \Delta^{(t)} \right\|^{1/2} \left\| \frac{1}{T} \sum_{t=1}^T F_\star^{(t)\top} F_\star^{(t)} \right\|^{1/2} + 2 \left\| \frac{1}{T} \sum_{t=1}^T E^{(t)\top} E^{(t)} \right\|^{1/2} \left\| \frac{1}{T} \sum_{t=1}^T F_\star^{(t)\top} F_\star^{(t)} \right\|^{1/2}$$

$$+ 2 \left\| \frac{1}{T} \sum_{t=1}^T \Delta^{(t)\top} \Delta^{(t)} \right\|^{1/2} \left\| \frac{1}{T} \sum_{t=1}^T E^{(t)\top} E^{(t)} \right\|^{1/2}.$$

Using Lemma A.6, we get the following upper bound on $\lambda_{\max} \left( \frac{1}{T} \sum_{t=1}^T \hat{F}^{(t)\top} \hat{F}^{(t)} \right)$.

**Lemma A.10** *Let*

$$N_1 \gtrsim \max \left\{ \gamma^4 \left( r + \log(T/\delta) \right), \frac{\lambda_{\max}^{\mathbf{F}}{}^{-1}}{T} \sum_{t=1}^T \frac{\sigma_w^{(t)2}(d_y + r + \log(T/\delta))}{\lambda_{\min}(\Sigma_x^{(t)})} \right\},$$

*and* $\operatorname{dist}(\hat{\Phi}, \Phi_\star) \le \frac{1}{5} \frac{\sqrt{\lambda_{\max}^{\mathbf{F}}}}{T} \sum_{t=1}^T \kappa\left( \Sigma_x^{(t)} \right)^{-1}$. *Then, with probability at least $1 - \delta$, we have*

$$\lambda_{\max} \left( \frac{1}{T} \sum_{t=1}^T \hat{F}^{(t)\top} \hat{F}^{(t)} \right) \le 3\lambda_{\max}^{\mathbf{F}}.$$

*Proof:* we see that it suffices to establish $\left\| \frac{1}{T} \sum_{t=1}^{T} \Delta^{(t)\top} \Delta^{(t)} \right\| \lesssim \left\| \frac{1}{T} \sum_{t=1}^{T} F_{\star}^{(t)\top} F_{\star}^{(t)} \right\| =: \lambda_{\max}^{\mathbf{F}}$

and $\left\| \frac{1}{T} \sum_{t=1}^{T} E^{(t)\top} E^{(t)} \right\| \lesssim \lambda_{\max}^{\mathbf{F}}$ in order to establish

$$\lambda_{\max} \left( \frac{1}{T} \sum_{t=1}^{T} \hat{F}^{(t)\top} \hat{F}^{(t)} \right) \lesssim \lambda_{\max}^{\mathbf{F}}.$$

We recall that

$$\Delta^{(t)} := \Phi_{\star} \left( I_{d_x} - \hat{\Phi}^{\top} \hat{\Phi} \right) X_{\mathcal{N}_1}^{(t)\top} Z_{\mathcal{N}_1}^{(t)} \left( \hat{\Sigma}_{z,\mathcal{N}_1}^{(t)} \right)^{-1}$$

$$E^{(t)} := W_{\mathcal{N}_1}^{(t)\top} Z_{\mathcal{N}_1}^{(t)} \left( \hat{\Sigma}_{z,\mathcal{N}_1}^{(t)} \right)^{-1},$$

which by Lemma A.6 admit the following high-probability bounds: let $|\mathcal{N}_1| := N_1 \gtrsim \gamma^4 \left( r + \log(1/\delta) \right)$. Then, with probability greater than $1 - \delta$, we have

$$\left\| \Phi_{\star} \mathcal{P}_{\hat{\Phi}}^{\perp} X_{\mathcal{N}_1}^{(t)\top} Z_{\mathcal{N}_1}^{(t)} \left( \hat{\Sigma}_{z,\mathcal{N}_1}^{(t)} \right)^{-1} \right\| \leq \frac{5}{4} \mathrm{dist}(\hat{\Phi}, \Phi_{\star}) \kappa \left( \Sigma_x^{(t)} \right)$$

$$\left\| W_{\mathcal{N}_1}^{(t)\top} Z_{\mathcal{N}_1}^{(t)} \left( \hat{\Sigma}_{z,\mathcal{N}_1}^{(t)} \right)^{-1} \right\| \lesssim \sigma_w^{(t)} \sqrt{\frac{d_y + r + \log(1/\delta)}{\lambda_{\min}(\Sigma_x^{(t)}) N_1}}.$$

Therefore, inverting the RHS' for a prescribed numerical constant factor of $\lambda_{\max}^{\mathbf{F}}$ yields the result. ∎

Lemma A.10 informs the maximum we may set the step-size. To now upper bound the contraction rate, we lower bound $\lambda_{\min}(\hat{F}^{(t)\top} \hat{F}^{(t)})$. We observe that the diagonal terms $\Delta^{(t)\top} \Delta^{(t)}$, $E^{(t)\top} E^{(t)}$ are pd, and thus can be ignored. We then observe that by Weyl's inequality (Horn and Johnson, 2012), we have

$$\lambda_{\min}(\hat{F}^{(t)\top} \hat{F}^{(t)}) \geq \lambda_{\min} \left( \hat{\Phi} \Phi_{\star}^{\top} F_{\star}^{(t)\top} F_{\star}^{(t)} \Phi_{\star} \hat{\Phi}^{\top} \right)$$

$$- \lambda_{\max} \left( \mathrm{Sym}(\Delta^{(t)\top} F_{\star}^{(t)} \Phi_{\star} \hat{\Phi}^{\top}) + \mathrm{Sym}(E^{(t)\top} F_{\star}^{(t)} \Phi_{\star} \hat{\Phi}^{\top}) + \mathrm{Sym}(\Delta^{(t)\top} E^{(t)}) \right).$$

Just as in the upper bound of $\lambda_{\max}(\hat{F}^{(t)\top} \hat{F}^{(t)})$, it suffices to upper bound the cross terms. Therefore, following the same proof as Lemma A.10, we have the analogous result.

**Lemma A.11** *Let*

$$N_1 \gtrsim \max \left\{ \gamma^4 \left( r + \log(1/\delta) \right), \frac{\lambda_{\max}^{\mathbf{F}}}{(\lambda_{\min}^{\mathbf{F}})^2} \frac{1}{T} \sum_{t=1}^{T} \frac{\sigma_w^{(t)2}(d_y + r + \log(1/\delta))}{\lambda_{\min}(\Sigma_x^{(t)})} \right\},$$

*and* $\mathrm{dist}(\hat{\Phi}, \Phi_{\star}) \leq \min\{ \frac{1}{30} \frac{\lambda_{\min}^{\mathbf{F}}}{\sqrt{\lambda_{\max}^{\mathbf{F}}}} \frac{1}{T} \sum_{t=1}^{T} \kappa \left( \Sigma_x^{(t)} \right)^{-1}, 1/2 \}$. *Then, with probability at least* $1 - \delta$, *we have*

$$\lambda_{\min} \left( \frac{1}{T} \sum_{t=1}^{T} \hat{F}^{(t)\top} \hat{F}^{(t)} \right) \geq \frac{1}{2} \lambda_{\min}^{\mathbf{F}}.$$

*Proof:* as in Lemma A.10, we invert the bounds from Lemma A.6 for our desired factors of $\lambda_{\min}^{\mathbf{F}}$. By our proposed burn-in, we have with probability at least $1 - \delta$

$$\lambda_{\max} \left( \mathrm{Sym}(\Delta^{(t)\top} F_{\star}^{(t)} \Phi_{\star} \hat{\Phi}^{\top}) + \mathrm{Sym}(E^{(t)\top} F_{\star}^{(t)} \Phi_{\star} \hat{\Phi}^{\top}) + \mathrm{Sym}(\Delta^{(t)\top} E^{(t)}) \right) \leq \frac{1}{4} \lambda_{\min}^{\mathbf{F}}.$$

The only additional factor we account for here is the lower bound on $\lambda_{\min} \left( \hat{\Phi} \Phi_{\star}^{\top} F_{\star}^{(t)\top} F_{\star}^{(t)} \Phi_{\star} \hat{\Phi}^{\top} \right)$. We have

$$\lambda_{\min} \left( \hat{\Phi} \Phi_{\star}^{\top} F_{\star}^{(t)\top} F_{\star}^{(t)} \Phi_{\star} \hat{\Phi}^{\top} \right) = \min_{\|x\|=1} x^{\top} \hat{\Phi} \Phi_{\star}^{\top} F_{\star}^{(t)\top} F_{\star}^{(t)} \Phi_{\star} \hat{\Phi}^{\top} x$$

$$\geq \lambda_{\min} \left( F_{\star}^{(t)\top} F_{\star}^{(t)} \right) \min_{\|x\|=1} x^{\top} \hat{\Phi} \Phi_{\star}^{\top} \Phi_{\star} \hat{\Phi}^{\top} x$$

$$= \lambda_{\min} \left( F_{\star}^{(t)\top} F_{\star}^{(t)} \right) \sigma_{\min}^2 \left( \Phi_{\star} \hat{\Phi}^{\top} \right).$$

To further lower bound $\sigma_{\min}^2\left(\Phi_\star\hat{\Phi}^\top\right)$, we observe that

$$\hat{\Phi}\hat{\Phi}^\top = \hat{\Phi}\left(\Phi_\star^\top\Phi_\star + \Phi_{\star,\perp}^\top\Phi_{\star,\perp}\right)\hat{\Phi}^\top$$

$$\implies 1 = \lambda_{\max}(\hat{\Phi}\hat{\Phi}^\top) \leq \lambda_{\min}\left(\hat{\Phi}\Phi_\star^\top\Phi_\star\hat{\Phi}^\top\right) + \lambda_{\max}\left(\hat{\Phi}\Phi_{\star,\perp}^\top\Phi_{\star,\perp}\hat{\Phi}^\top\right) \quad \text{Weyl's inequality}$$

$$\implies \sigma_{\min}^2\left(\Phi_\star\hat{\Phi}^\top\right) \geq 1 - \left\|\Phi_{\star,\perp}\hat{\Phi}^\top\right\|^2 =: 1 - \operatorname{dist}(\hat{\Phi},\Phi_\star)^2.$$

Under our assumption that $\operatorname{dist}(\hat{\Phi},\Phi_\star) \leq \frac{1}{2}$, we can piece together this bound with the bound on the cross terms to yield with probability at least $1-\delta$

$$\lambda_{\min}(\hat{F}^{(t)\top}\hat{F}^{(t)}) \geq \left(\frac{3}{4} - \frac{1}{4}\right)\lambda_{\min}(F_\star^{(t)\top}F_\star^{(t)}) = \frac{1}{2}\lambda_{\min}(F_\star^{(t)\top}F_\star^{(t)}),$$

which completes the proof. ∎

Piecing together Lemma A.10 and Lemma A.11, we get the following bound on the contraction factor:

**Lemma A.12** *Let*

$$N_1 \gtrsim \max\left\{\gamma^4\left(r + \log(1/\delta)\right), \frac{\lambda_{\max}^{\mathbf{F}}}{(\lambda_{\min}^{\mathbf{F}})^2}\frac{1}{T}\sum_{t=1}^T \frac{\sigma_w^{(t)2}(d_y + r + \log(1/\delta))}{\lambda_{\min}(\Sigma_x^{(t)})}\right\},$$

*and* $\operatorname{dist}(\hat{\Phi},\Phi_\star) \leq \min\{\frac{1}{30}\frac{\lambda_{\min}^{\mathbf{F}}}{\sqrt{\lambda_{\max}^{\mathbf{F}}}}\frac{1}{T}\sum_{t=1}^T \kappa\left(\Sigma_x^{(t)}\right)^{-1}, 1/2\}$. *Then, for step-size satisfying* $\eta \leq \frac{1}{3\lambda_{\max}^{\mathbf{F}}}$, *with probability at least* $1-\delta$, *we have*

$$\left\|I_{d_x} - \eta\frac{1}{T}\sum_{t=1}^T \hat{F}^{(t)\top}\hat{F}^{(t)}\right\| \leq \left(1 - \frac{\eta}{2}\lambda_{\min}^{\mathbf{F}}\right).$$

### A.3 The Non-IID Setting

To extend our analysis to the non-iid setting, we first instantiate our covariates as $\beta$-mixing stationary processes (Yu, 1994; Kuznetsov and Mohri, 2017).

**Assumption A.1 (Geometric mixing)** *For each $h$, assume the process $\{x_i^{(t)}\}_{t\geq 1}$ is a mean-zero stationary $\beta$-mixing process, with stationary covariance $\Sigma_x^{(t)}$ and $\beta(k) := \Gamma\mu^k$.*

We note that exact stationarity is unnecessary as long as the marginal distributions converge to stationarity sufficiently fast; however, we assume exact stationarity for convenience. We now invoke the blocking technique on each trajectory, where each trajectory is subsampled into $a$ trajectories of length $m$ (we assume $N = ma$ for notational convenience). We may then apply the analysis of the iid setting on a deflated dataset of $Tm$ data points *drawn from the respective stationary distributions* to yield:

**Proposition A.3** *Let* $x_i^{(t)} \sim \nu_\infty^{(t)}$ *for each* $t \in [T]$, $i \in [N]$. *Assume* $|\mathcal{N}_1| \gtrsim \max\left\{\gamma^2, \frac{\sigma_w^{(t)2}}{\lambda_{\min}(\Sigma_x^{(t)})}\frac{\|F_\star\|^2}{\sigma_{\min}(F_\star^{(t)})^4}\right\}(\max\{d_y, r\} + \log(H/\delta))$, *and* $\left\|\hat{\Phi}\Phi_{\star,\perp}^\top\right\| \leq \min\left\{\frac{3}{16}\frac{1}{\kappa\left(\Sigma_x^{(t)}\right)+1}\frac{\sigma_{\min}(F_\star^{(t)})^2}{\|F_\star\|}, 1/2\right\}$. *Let* $\eta \leq \min_t 1/4\|F_\star^{(t)}\|^2$. *There exists numerical constant* $C > 0$ *such that the following holds: with probability at least* $1-\delta$,

$$\left\|\hat{\Phi}_+\Phi_{\star,\perp}^\top\right\| \leq \left(1 - \frac{3\eta}{16}\frac{1}{T}\sum_{t=1}^T \sigma_{\min}(F_\star^{(t)})^2\right)^{1/2}\left\|\hat{\Phi}\Phi_{\star,\perp}^\top\right\|$$

$$+ \max_t C\frac{\eta\sigma_w^{(t)}\|F_\star^{(t)}\|}{\sqrt{|\mathcal{N}_2|m}}\sqrt{\frac{d_x + \log(H/\delta)}{\lambda_{\min}(\Sigma_x^{(t)})}}\log\left(\frac{d_y + r}{\delta}\right).$$

Now applying Lemma A.4, setting $g(\cdot)$ as the indicator function for the burn-in requirement and the final descent bound, we have for all $j = 1, \ldots, a$.

$$\left| \mathbb{E} \left[ g \left( \left\{ X_\infty^{(t),Nm} \right\}_{t \in [T]} \right) \right] - \mathbb{E} \left[ g \left\{ X_{(j)}^{(t),NT} \right\}_{t \in [T]} \right] \right| \leq m\beta(a) \leq \delta'$$

Setting $\delta' = \delta/a$ and union bounding over each $j = 1, \ldots, a$, we may invert $N\beta(a) = \delta$ to find $a := \log\left(\frac{\Gamma N}{\delta}\right) / \log\left(\frac{1}{\mu}\right)$. This yields the final descent guarantee.

**Theorem A.1** *Let Assumption A.1 hold for the processes $\{x_i^{(t)}\}_{t=1}^T$ for each $i \in [N]$, $t \in [T]$. Define $m := \frac{\log(1/\mu)T}{\log\left(\frac{\Gamma T}{\delta}\right)}$. Assume $|\mathcal{N}_1|m \gtrsim \max\left\{\gamma^2, \frac{\sigma_w^{(t)2}}{\lambda_{\min}(\Sigma_x^{(t)})} \frac{\|F_\star\|^2}{\sigma_{\min}(F_\star^{(t)})^4}\right\} (\max\{d_y, r\} + \log(H/\delta))$, and $\left\|\hat{\Phi}\Phi_{\star,\perp}^\top\right\| \leq \min\left\{\frac{3}{16} \frac{1}{\kappa(\Sigma_x^{(t)})+1} \frac{\sigma_{\min}(F_\star^{(t)})^2}{\|F_\star\|}, 1/2\right\}$. Let $\eta \leq \min_t 1/4\|F_\star^{(t)}\|^2$. There exists numerical constant $C > 0$ such that the following holds: with probability at least $1 - \delta$,*

$$\left\|\hat{\Phi}_+\Phi_{\star,\perp}^\top\right\| \leq \left(1 - \frac{3\eta}{16}\frac{1}{T}\sum_{t=1}^T \sigma_{\min}(F_\star^{(t)})^2\right)^{1/2} \left\|\hat{\Phi}\Phi_{\star,\perp}^\top\right\|$$

$$+ \max_t C\frac{\eta\sigma_w^{(t)}\|F_\star^{(t)}\|}{\sqrt{|\mathcal{N}_2|m}} \sqrt{\frac{d_x + \log(T/\delta)}{\lambda_{\min}(\Sigma_x^{(t)})}} \log\left(\frac{d_y + r}{\delta}\right)$$

$$= \left(1 - \frac{3\eta}{16}\frac{1}{T}\sum_{t=1}^T \sigma_{\min}(F_\star^{(t)})^2\right)^{1/2} \left\|\hat{\Phi}\Phi_{\star,\perp}^\top\right\|$$

$$+ \tilde{\mathcal{O}}\left(\max_t \frac{\eta\sigma_w^{(t)}\|F_\star^{(t)}\|}{\sqrt{|\mathcal{N}_2|T}} \sqrt{\frac{d_x + \log(T/\delta)}{\lambda_{\min}(\Sigma_x^{(t)})}} \log\left(\frac{d_y + r}{\delta}\right)\right).$$

## A.4 CONVERTING TO SAMPLE COMPLEXITY BOUNDS

To highlight the importance of the task scaling $T$ in our descent guarantees, we demonstrate how to convert general descent lemmas to sample complexity guarantees.

**Lemma A.13** *For a sequence of positive integers $\{M_k\}_{k \geq 1} \subset \mathbb{N}$, define $\{d_k\}_{k \geq 1} \subset \mathbb{R}_+$ as a sequence of non-negative real numbers dependent on $\{M_k\}$ that satisfy the relation*

$$d_{k+1} \leq \rho \cdot d_k + \frac{C}{M_k},$$

*for some $\rho \in (0,1)$ and $C > 0$. Let $d_0 = \tau$. Given a positive integer $M$, we may partition $M = \sum_{k=1}^K M_k$, where*

$$K := \left\lfloor \frac{1}{2}\log\left(\frac{2}{1+\rho}\right)^{-1} \frac{M\tau^2}{C}\left(\frac{1-\rho}{2}\right)^3 + 1 \right\rfloor,$$

*such that the following guarantee holds on $d_K$:*

$$d_K \leq \tau\sqrt{\frac{2C}{M}\left(\frac{2}{1-\rho}\right)^3}.$$

The proof of Lemma A.13 follows by setting each $M_k$ such that $\rho \cdot d_k + \frac{C}{M_k} = \left(\frac{1+\rho}{2}\right)d_k$, and setting $K$ as the maximal $K$ such that $\sum_{k=1}^K M_k \leq M$. Evaluating $d_K \leq \tau\left(\frac{1+\rho}{2}\right)^K$ yields the result. For convenience, we do not consider burn-in times $M_k \geq M_0 \; \forall k$ or pseudo-linear dependence $\frac{C \text{polylog}(M_k)}{M_k}$. However, these will only lead to inflating $d_K$ by a polylog$(M)$ factor.

In essence, Lemma A.13 demonstrates how a fixed offline dataset of size $M$ can be partitioned into independent blocks of increasing size such that the final iterate satisfies an approximate ERM bound scaling as $\frac{1}{\sqrt{M}}$, inflated by a function of the contraction rate $\rho$. Instantiating Lemma A.13 with the problem parameters of Theorem 3.1 yields Corollary 3.1.

### A.4.1 Near-ERM Transfer Learning

An important consequence of Lemma A.13 (thus Corollary 3.1) is that near-ERM parameter recovery bounds can be extracted. In particular, given some $h \in [H+1]$, for a given representation $\hat{\Phi}$, and the least squares weights $\hat{F}^{(t)}$ computed with respect to some independent dataset of size $NT$,

$$
\begin{aligned}
\left\| \hat{M}^{(t)} - M_\star^{(t)} \right\|_F^2 &= \left\| \hat{F}^{(t)} \hat{\Phi} - F_\star^{(t)} \Phi_\star \right\|_F^2 \\
&= \left\| \hat{F}^{(t)} \hat{\Phi} \begin{bmatrix} \hat{\Phi}^\top & \hat{\Phi}_\perp^\top \end{bmatrix} - F_\star^{(t)} \Phi_\star \begin{bmatrix} \hat{\Phi}^\top & \hat{\Phi}_\perp^\top \end{bmatrix} \right\|_F^2 \\
&= \left\| \begin{bmatrix} \hat{F}^{(t)} - F_\star^{(t)} \Phi_\star \hat{\Phi}^\top & -F_\star^{(t)} \Phi_\star \hat{\Phi}_\perp^\top \end{bmatrix} \right\|_F^2 \\
&= \left\| \hat{F}^{(t)} - F_\star^{(t)} \Phi_\star \hat{\Phi}^\top \right\|_F^2 + \left\| F_\star^{(t)} \Phi_\star \hat{\Phi}_\perp^\top \right\|_F^2 \\
&\leq 2 \left\| F_\star^{(t)} \Phi_\star \left( I_{d_x} - \hat{\Phi}^\top \hat{\Phi} \right) X^{(t)\top} Z^{(t)} \left( \hat{\Sigma}_{z,\mathcal{N}_1}^{(t)} \right)^{-1} \right\|_F^2 \\
&\quad + 2 \left\| W^{(t)\top} Z^{(t)} \left( \hat{\Sigma}_{z,\mathcal{N}_1}^{(t)} \right)^{-1} \right\|_F^2 + \left\| F_\star^{(t)} \right\|^2 \operatorname{dist}(\hat{\Phi}, \Phi_\star)^2 \\
&\lesssim \left\| F_\star^{(t)} \right\|^2 \kappa \left( \Sigma_x^{(t)} \right) \operatorname{dist}(\hat{\Phi}, \Phi_\star)^2 + \sigma_w^{(t)2} \frac{d_y r + \log(1/\delta)}{\lambda_{\min}(\Sigma_x^{(t)}) NT} \quad \text{w.p. } \geq 1 - \delta,
\end{aligned}
$$

where the last line follows from applying Lemma A.6. We observe that the parameter error nicely decomposes into a term quadratic in $\operatorname{dist}(\hat{\Phi}, \Phi_\star)$ and least squares fine-tuning error scaling with $\frac{1}{NT}$. For a fixed dataset of size $NT$, one can crudely set aside $\Theta(N)$ samples for each task, and use the rest of the $\Theta(N)$ samples to compute $\hat{\Phi}$. Invoking Corollary 3.1 and using the set-aside $\Theta(N)$ samples to compute $\hat{F}^{(t)}$ conditioned on $\hat{\Phi}$, we recover the near-ERM high probability generalization bound on the parameter error

$$
\left\| \hat{M}^{(t)} - M_\star^{(t)} \right\|_F^2 \leq \tilde{O} \left( \|F_\star^{(t)}\|^2 \kappa \left( \Sigma_x^{(t)} \right) C(\rho) \frac{\max_t \sigma_w^{(t)2} d_x r}{NT} + \frac{\sigma_w^{(t)2} d_y r}{\lambda_{\min}(\Sigma_x^{(t)}) N} \right).
$$

## B  Case Study: Linear Dynamical Systems

To understand the importance of permitting non-isotropy and sequential dependence in multi-task data, we consider the fundamental setting of linear systems, which has served as a staple testbed for statistical and algorithmic analysis in recent years, since it lends itself to non-trivial yet tractable *continuous* reinforcement learning problems (see e.g., (Hu et al., 2023; Krauth et al., 2019; Tu and Recht, 2018; 2019; Recht, 2019; Fazel et al., 2018)), as well as (online) statistical learning problems with temporally dependent covariates (Abbasi-Yadkori et al., 2011; Abbasi-Yadkori, 2013; Simchowitz and Foster, 2020; Dean et al., 2018; 2020; Agarwal et al., 2019a;b; Ziemann et al., 2022; Ziemann and Sandberg, 2022; Lee et al., 2023; Simchowitz et al., 2018) (see (Tsiamis et al., 2022) for a tutorial and literature review). In particular for our purposes, the dependence of contiguous covariates in a linear system is intricately connected to its *stability properties* (Simchowitz et al., 2018; Jedra and Proutiere, 2020; Tu et al., 2022), such that we may instantiate the guarantees of DFW for non-iid data in an interpretable manner.

The standard state-space linear system set-up admits the form

$$
\begin{aligned}
s[t+1] &= A^{(h)} s[t] + B^{(h)} u[t] + w[t] \\
w[t] &\overset{\text{i.i.d.}}{\sim} \mathcal{N}(0, \Sigma_w^{(h)}), \quad s[0] \sim \mathcal{N}(0, \Sigma_0^{(h)}),
\end{aligned}
\tag{27}
$$

where we preemptively index possibly task-specific quantities. We consider the following two common linear system settings: system identification and imitation learning.

## B.1 LINEAR SYSTEM IDENTIFICATION

In linear system identification, the aim is to estimate the system matrices $(A^{(h)}, B^{(h)})$ given state and input measurements $s_t, u_t$. In particular, we may cast the sysID problem as the following regression:

$$s[t+1] = \begin{bmatrix} A^{(h)} & B^{(h)} \end{bmatrix} \begin{bmatrix} s[t] \\ u[t] \end{bmatrix} + w[t].$$

It is customary to consider exploratory signals that are iid zero-mean Gaussian random vectors $u[t] \overset{\text{i.i.d.}}{\sim} \mathcal{N}(0, \Sigma_u^{(h)})$ (Simchowitz et al., 2018; Simchowitz and Foster, 2020; Tsiamis et al., 2022). In the stable system case, $\rho(A^{(h)}) < 1$, we can therefore evaluate the covariance of the *stationary* distribution of states $s[t]$ induced by exploratory signal $u[t] \overset{\text{i.i.d.}}{\sim} \mathcal{N}(0, \Sigma_u^{(h)})$ by plugging in (27) into the following equation

$$\begin{aligned}
\mathbb{E}_{u,w}[s[t]s[t]^\top] &= \mathbb{E}_{u,w}\left[s[t+1]s[t+1]^\top\right] \\
&= A^{(h)}\mathbb{E}_{u,w}\left[s[t]s[t]^\top\right]A^{(h)^\top} + B^{(h)}\mathbb{E}\left[u[t]u[t]^\top\right]B^{(h)^\top} + \mathbb{E}\left[w[t]w[t]^\top\right] \\
&= A^{(h)}\mathbb{E}_{u,w}\left[s[t]s[t]^\top\right]A^{(h)^\top} + B^{(h)}\Sigma_u^{(h)}B^{(h)^\top} + \Sigma_w^{(h)}
\end{aligned}$$

Therefore, evaluating the stationary state covariance $\Sigma_s^{(h)} := \mathbb{E}\left[s[\infty]s[\infty]^\top\right]$ amounts to solving the Discrete Lyapunov Equation (`dlyap`):

$$\Sigma_s^{(h)} := A^{(h)}\Sigma_s^{(h)}A^{(h)^\top} + B^{(h)}\Sigma_u^{(h)}B^{(h)^\top} + \Sigma_w^{(h)}.$$

In the notation introduced earlier in the paper, casting $y[t] \leftarrow s[t+1]$, $x[t] \leftarrow \begin{bmatrix} s[t] \\ u[t] \end{bmatrix}$, $M_\star^{(h)} \leftarrow \begin{bmatrix} A^{(h)} & B^{(h)} \end{bmatrix}$, we may instantiate multi-task linear system identification as a non-iid, non-isotropic linear operator recovery problem.

**Definition B.1** *Let the initial state covariance be the stationary covariance $\Sigma_0^{(h)} = \Sigma_s^{(h)}$, such that the covariance of the marginal covariate distribution satisfies*

$$\Sigma_x^{(h)} := \mathbb{E}\left[x[t]x[t]^\top\right] = \begin{bmatrix} \Sigma_s^{(h)} & 0 \\ 0 & \Sigma_u^{(h)} \end{bmatrix}, \text{ for all } t \geq 0.$$

We make the above standard definition for the initial state distribution for convenience, as it ensures the marginal distributions of each state are identical. We note, however, given a different initial state distribution, the marginal state distribution converges exponentially quickly to stationarity, thus accumulating only a negligible factor to the final rates. We then make the following system assumptions to instantiate our representation learning guarantees.

**Assumption B.1** *We assume that for any task $h$ the following hold:*

1. *The operators share a rowspace $M_\star^{(h)} := \begin{bmatrix} A^{(h)} & B^{(h)} \end{bmatrix} = F_\star^{(h)}\Phi_\star$, $F_\star^{(h)} \in \mathbb{R}^{d_s \times r}$, $\Phi_\star \in \mathbb{R}^{r \times (d_s + d_u)}$.*

2. *The state matrices have uniformly bounded spectral radii $\rho(A^{(h)}) < \mu < 1$. Subsequently, we assume there exists a constant $\Gamma' > 0$ that satisfies*

$$\|A^{(h)^k}\|_2 \leq \Gamma'\mu^k, \text{ for all } k \geq 0.$$

   *The existence of a uniform $\Gamma'$ is guaranteed by Gelfand's Formula (Horn and Johnson, 2012), and quantitative bounds may be found in, e.g., Goldenshluger and Zeevi (2001); Tu and Recht (2018).*

The first assumption is satisfied, for example, when $A^{(h)} = P_\star^{(h)}U_\star$ and $B^{(h)} = Q_\star^{(h)}V_\star$ individually admit low-rank decompositions. The second assumption translates to a quantitative bound on the mixing time of the covariates $x[t]$ by adapting a result from Tu and Recht (2018).

**Proposition B.1 (Adapted from Tu and Recht (2018, Prop. 3.1))** *For each h, let the dynamics for the linear system evolve as described in (27). Let Assumption B.1 hold with constants $\Gamma'$, $\rho$. Define $\mathbb{P}_{s[k]\sim\nu_k}[\,\cdot\mid s_0 = s]$ as the conditional distribution of states $s[k]$ given initial condition $s_0 = s$. We have for any $k \geq 0$ and initial state distribution $\nu_0$,*

$$\mathbb{E}_{s\sim\nu_0}\left[\left\|\mathbb{P}_{s[k]}[\,\cdot\mid s_0 = s] - \mathbb{P}_{s[k]}\right\|_{\mathrm{tv}}\right] \leq \frac{\Gamma'}{2}\sqrt{\mathbb{E}_{\nu_0}[\|s[0]\|^2] + \frac{\|\Sigma^{-1}\|_*}{1-\mu^2}} \cdot \mu^k, \qquad (28)$$

*where $\|\cdot\|_*$ indicates the nuclear norm (Horn and Johnson, 2012), and $\Sigma := B^{(h)}\Sigma_u^{(h)}B^{(h)^\top} + \Sigma_w^{(h)}$.*

We note that by the independence of control inputs $u[t]$, we have trivially that the total variation distance between the conditional and marginal distributions of covariates $x[t]$ is the same as that of the states $s[t]$.

$$\left\|\mathbb{P}_{s[k]}[\,\cdot\mid s_0 = s] - \mathbb{P}_{s[k]}\right\|_{\mathrm{tv}} = \left\|\mathbb{P}_{x[k]}[\,\cdot\mid s_0 = s] - \mathbb{P}_{x[k]}\right\|_{\mathrm{tv}}$$

Since by construction the marginal distribution of states is identically $\mathcal{N}(0, \Sigma_s^{(h)})$, applying Proposition B.1 to $s[t], s[t+k]$ for any $t, k$, we get the following quantitative bound on the mixing-time of the covariates $x[t] = \begin{bmatrix} s[t]^\top & u[t]^\top \end{bmatrix}^\top$.

**Lemma B.1** *Following Definition B.1 and Assumption B.1, the covariate process $\left\{x^{(h)}[t]\right\}_{t\geq0}$ is a mean-zero, stationary, geometrically $\beta$-mixing process with covariance $\Sigma_x^{(h)} = \begin{bmatrix} \Sigma_s^{(h)} & 0 \\ 0 & \Sigma_u^{(h)} \end{bmatrix}$, where $\Sigma_s^{(h)} = \mathrm{dlyap}(A^{(h)}, B^{(h)}\Sigma_u^{(h)}B^{(h)} + \Sigma_w^{(h)})$, and mixing-time bounded by*

$$\beta(k) = \Gamma\mu^k, \quad \text{where}$$

$$\Gamma := \frac{\Gamma'}{2}\sqrt{\mathrm{Tr}\left(\Sigma_s^{(h)}\right) + \frac{\|\Sigma^{-1}\|_*}{1-\mu^2}}, \ \Sigma := B^{(h)}\Sigma_u^{(h)}B^{(h)^\top} + \Sigma_w^{(h)}. \qquad (29)$$

Thus, instantiating Lemma B.1 in Theorem A.1 gives us guarantees of `DFW` applied to multi-task linear system identification.

## B.2 IMITATION LEARNING

In linear (state-feedback) imitation learning (IL), the aim is to estimate linear state-feedback controllers $K^{(h)} \in \mathbb{R}^{d_u \times d_x}$ from (noisy) state-input pairs $\{(s[t], u[t])\}_{t\geq0}$ induced by unknown expert controllers $K_\star^{(h)}$. In particular, we assume the expert control inputs are generated as

$$u[t] = K_\star^{(h)}s[t] + z[t], \quad z[t] \overset{\text{i.i.d.}}{\sim} \mathcal{N}(0, \Sigma_z^{(h)}),$$

which we observe lends itself naturally as a linear regression, casting $y[t] \leftarrow u[t]$, $x[t] \leftarrow s[t]$, $M_\star^{(h)} \leftarrow K_\star^{(h)}$. Plugging the expert control inputs into the dynamics (27) yields that the states/covariates evolve as

$$s[t+1] = A^{(h)}s[t] + B^{(h)}\left(K_\star^{(h)}s[t] + z[t]\right) + w[t]$$

$$= \left(A^{(h)} + B^{(h)}K_\star^{(h)}\right)s[t] + Bz[t] + w[t].$$

We make the natural assumption that the expert controller $K_\star^{(h)}$ stabilizes the system, i.e. the spectral radius of the closed-loop dynamics has spectral radius strictly less than 1: $\rho\left(A^{(h)} + B^{(h)}K_\star^{(h)}\right) < 1$. As such, similar to the linear sysID setting, we may plug the above dynamics into the stationarity equation to yield the stationary covariance:

$$\mathbb{E}[s[t]s[t]^\top] = \mathbb{E}\left[s[t+1]s[t+1]^\top\right]$$

$$= \left(A^{(h)} + B^{(h)}K_\star^{(h)}\right)\mathbb{E}[s[t]s[t]^\top]\left(A^{(h)} + B^{(h)}K_\star^{(h)}\right)^\top + B^{(h)}\Sigma_z^{(h)}B^{(h)^\top} + \Sigma_w^{(h)}$$

$$\implies \Sigma_s^{(h)} = \mathrm{dlyap}\left(A^{(h)} + B^{(h)}K_\star^{(h)}, B^{(h)}\Sigma_z^{(h)}B^{(h)^\top} + \Sigma_w^{(h)}\right).$$

Analogously to linear sysID, we make the following assumptions.

**Assumption B.2** *We assume that for any task $h$ the following hold:*

1. *The initial state covariance is set to the stationary covariance $\Sigma_0^{(h)} = \Sigma_s^{(h)}$, such that the marginal covariate distributions satisfy*

$$\mathbb{E}\left[x[t]x[t]^\top\right] = \Sigma_s^{(h)} =: \Sigma_x^{(h)}, \text{ for all } t \geq 0.$$

2. *The controllers share a rowspace $M_\star^{(h)} \equiv K_\star^{(h)} = F_\star^{(t)}\Phi_\star$, $F_\star^{(t)} \in \mathbb{R}^{d_u \times r}$, $\Phi_\star \in \mathbb{R}^{r \times d_s}$.*

3. *The closed-loop dynamics have uniformly bounded spectral radii $\rho\left(A^{(h)} + B^{(h)}K_\star^{(h)}\right) < \mu < 1$. Subsequently, we assume there exists a constant $\Gamma' > 0$ that satisfies*

$$\left\|\left(A^{(h)} + B^{(h)}K_\star^{(h)}\right)^k\right\|_2 \leq \Gamma'\mu^k.$$

*The existence of uniform $\Gamma'$ is guaranteed by Gelfand's Formula (Horn and Johnson, 2012).*

By using a result almost identical to Proposition B.1, we yield the following quantitative bound on the mixing time of covariates generated by stabilizing expert controllers.

**Lemma B.2** *Following Assumption B.2, the covariate process $\left\{x^{(h)}[t]\right\}_{t \geq 0}$ is a mean-zero, stationary, geometrically $\beta$-mixing process with covariance $\Sigma_x^{(h)} = \Sigma_s^{(h)}$, where $\Sigma_s^{(h)} = \text{dlyap}\left(A^{(h)} + B^{(h)}K_\star^{(h)}, B^{(h)}\Sigma_z^{(h)}B^{(h)\top} + \Sigma_w^{(h)}\right)$, and mixing-time bounded by*

$$\beta(k) = \Gamma\mu^k, \quad where$$
$$\Gamma := \frac{\Gamma'}{2}\sqrt{\text{Tr}\left(\Sigma_s^{(h)}\right) + \frac{\|\Sigma^{-1}\|_*}{1 - \mu^2}}, \quad \Sigma := B^{(h)}\Sigma_z^{(h)}B^{(h)\top} + \Sigma_w^{(h)}. \tag{30}$$

Thus, instantiating Lemma B.1 in Theorem A.1 gives us guarantees of `DFW` applied to multi-task linear imitation learning.

## C    ADDITIONAL NUMERICAL EXPERIMENTS AND DETAILS

We present additional numerical experiments to demonstrate the effectiveness of `DFW` (Algorithm 1) and provide a more detailed explanation of the task-generating process for constructing random operators in linear regression and system identification examples. Furthermore, we introduce an additional setting, imitation learning, to illustrate the advantages of collaborative learning across tasks in learning a linear quadratic regulator by leveraging expert data to compute a shared common representation across all tasks. In this latter setting, we also emphasize the importance of feature whitening when dealing with non-i.i.d. and non-isotropic data.

- **Random rotation:** For all the numerical experiments presented in this paper, the application of a random rotation around the identity is employed for both task-specific weight generation and the initialization of the representation. This random rotation is defined as $R_{\text{rot}} = \exp(\tilde{L})$, where $\tilde{L} = \frac{L - L^\top}{2}$ and $L = \gamma S$. Here, $S$ is a random matrix with entries drawn from a standard normal distribution, $d_l$ is the corresponding dimension of the high-dimensional latent space, and $\gamma$ corresponds to the scale of the rotation. We set $\gamma = 0.01$ for generating different task weights and $\gamma = 1$ for initializing the representation.

- **Step-sizes:** The step-size $\eta$ used to update the common representation is carefully selected to ensure a fair comparison between Algorithm 1 and the vanilla alternating minimization-descent approach employed in `FedRep` Collins et al. (2021). For example, to obtain the results depicted in Figure **??**, we set $\eta = 7, 5 \times 10^{-3}$, while for the alternating minimization-descent approach, which demonstrates better performance with a smaller step-size, we set $\eta = 5 \times 10^{-5}$ to achieve the results presented in Figure **??**. In Figure 1a, both the single-task and multi-task implementations of Algorithm 1 adopt $\eta = 7, 5 \times 10^{-3}$, whereas the vanilla

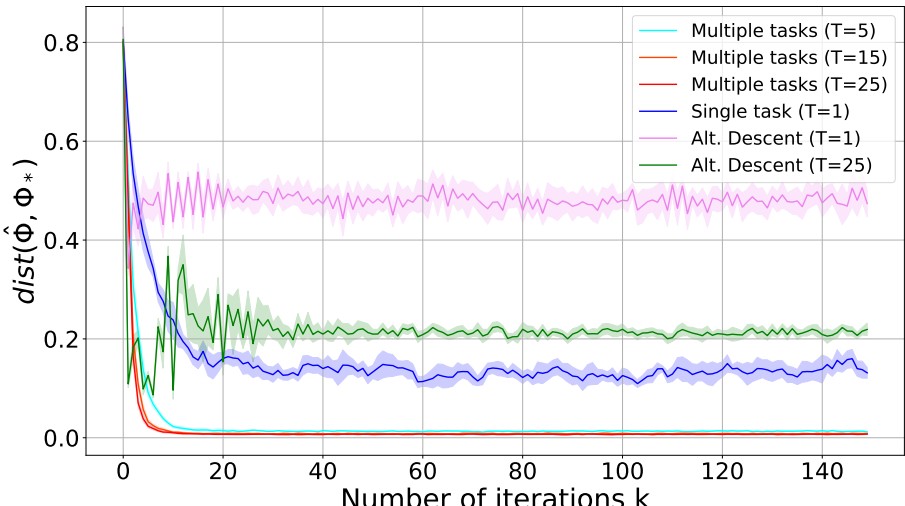

**Figure 2:** We plot the subspace distance between the current and ground truth representation with respect to the number of iterations, comparing between the single and multiple-task settings of Algorithm 1 and the multi-task `FedRep` for the IID linear regression with random covariance. We observe performance improvement and variance reduction for multi-task `DFW` as predicted.

alternating minimization-descent approach uses $\eta = 7.5 \times 10^{-3}$ for a fair comparison. Similarly, in Figure 1c, both the single-task and multi-task versions of Algorithm 1 use $\eta = 1 \times 10^{-1}$, while the vanilla alternating minimization-descent approach utilizes $\eta = 2 \times 10^{-3}$.

## C.1 Linear Regression with IID and Non-isotropic Data

Continuing our experiments for the linear regression problem, this time with different random linear operators as illustrated in Figure 1c, we present the results for an extended range of tasks using Algorithm 1 and the alternating minimization-descent approach (`FedRep` Collins et al. (2021)). In this analysis, we utilize the same specific parameters as discussed in §4. Additionally, we set the step-size $\eta = 7.5 \times 10^{-3}$ for both the single-task and multi-task implementations of Algorithm 1, and $\eta = 7.5 \times 10^{-5}$ for both the single-task and multi-task alternating minimization-descent.

Figure 2 presents a comparison of the performance between Algorithm 1 and the vanilla alternating minimization approach in both single and multi-task settings. In line with our theoretical results, the figure demonstrates that as the number of tasks $T$ increases, the error between the current representation and the ground truth representation significantly diminishes. In the specific case of linear regression with iid and non-isotropic data, this figure emphasizes that a small number of tasks ($T = 5$), is sufficient to achieve a low error in computing a shared representation across the tasks. Furthermore, the depicted figure reveals that while the multi-task alternating descent algorithm outperforms the single-task case, it is worth noting that this algorithm remains sub-optimal and is unable to surpass the limitation imposed by the presence of bias in the non-isotropic data. Despite its improved performance, the multi-task alternating descent algorithm still encounters challenges in overcoming the inherent noise barrier.

## C.2 System Identification

Building upon the results presented in §4, we conduct an extended experiment involving a larger range of tasks while maintaining the parameters specified in §4.2. Specifically, we generate distinct random

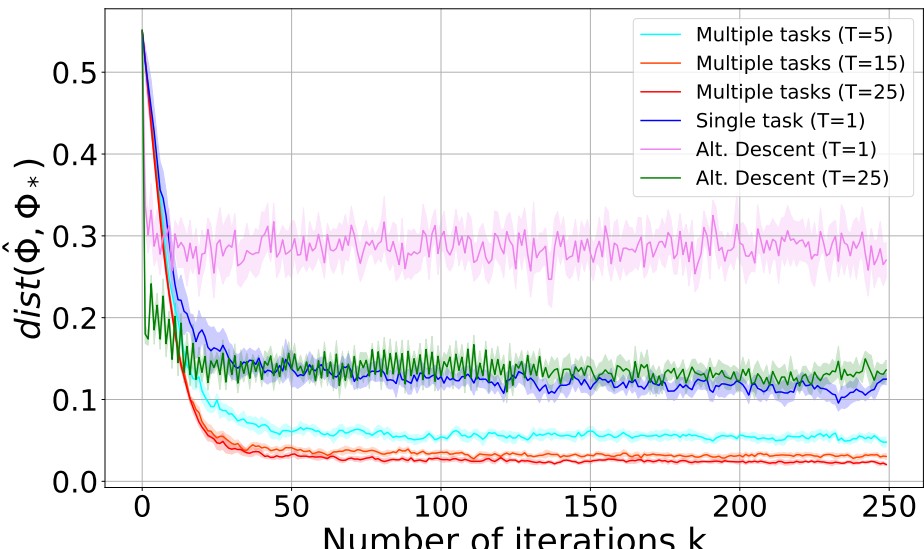

**Figure 3:** We plot the subspace distance between the current and ground truth representation with respect to the number of iterations, comparing between the single and multiple-task settings of Algorithm 1 multi-task `FedRep` for the linear system identification with random covariance. We observe performance improvement and variance reduction for multi-task `DFW` as predicted.

operators different from those utilized to obtain the results illustrated in Figure 1c. In this current analysis, we present the outcomes for the expanded range of tasks using Algorithm 1 and compare them to the single-task and multi-task vanilla alternating minimization-descent algorithms. The step-size $\eta$ is set to $1 \times 10^{-1}$ for both the single-task and multi-task implementations of Algorithm 1, while for the single-task and multi-task vanilla alternating minimization-descent algorithms, we set $\eta$ to $2 \times 10^{-3}$.

In alignment with our main theoretical findings, Figure 3 provides compelling evidence regarding the advantages of the proposed algorithm (Algorithm 1) compared to the vanilla alternating descent approach when computing a shared representation for all tasks. Consistent with the trend observed in Figure 2 for the linear regression problem, Figure 3 illustrates a significant reduction in the error between the current representation and the ground truth representation as the number of tasks increases. Additionally, it is noteworthy that while the multi-task alternating descent outperforms the single-task scenario, the single-task variant of Algorithm 1 achieves even better results. This observation underscores the importance of incorporating de-biasing and feature-whitening techniques when dealing with non-iid and non-isotropic data.

### C.3 Imitation Learning

Our focus now turns to the problem of learning a linear quadratic regulator (LQR) controller, denoted as $K^{(T+1)} = F_\star^{(T+1)} \Phi_\star$, by imitating the behavior of $T$ expert controllers $K^{(1)}, K^{(2)}, \ldots, K^{(T)}$. These controllers share a common low-rank representation and can be decomposed into the form $K^{(t)} = F_\star^{(t)} \Phi_\star$, where $F_\star^{(t)}$ represents the task-specific weight and $\Phi_\star$ corresponds to the common representation across all tasks. To achieve this, we exploit Algorithm 1 to compute a shared low-rank representation for all tasks by leveraging data obtained from the expert controllers. Within this context, we consider a discrete-time linear time-invariant dynamical system as follows:

$$x^{(t)}[i+1] = Ax^{(t)}[i] + Bu^{(t)}[i], \ \ i = 0, 1, \ldots, N-1,$$

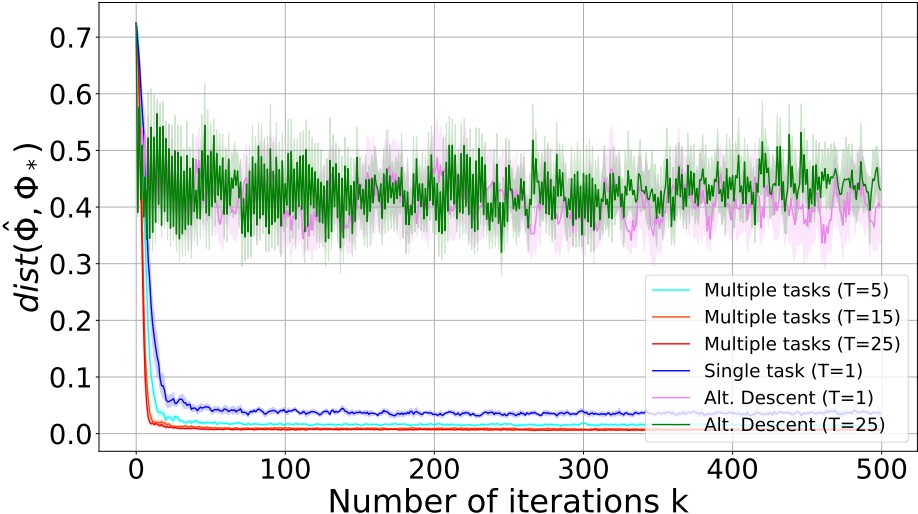

**Figure 4:** We plot the subspace distance between the current and ground truth representation with respect to the number of iterations, comparing between the single and multiple-task settings of Algorithm 1 and the multi-task `FedRep` for the imitation learning with random covariance. We observe performance improvement and variance reduction for multi-task `DFW` as predicted.

with $n_x = 4$ states and $n_u = 4$ inputs, for all $t \in [T+1]$, where $u^{(t)}[i] = K^{(t)}x^{(t)}[i] + z^{(t)}[i]$, with $z^{(t)}[i] \sim \mathcal{N}(0, I_{n_u})$ being the input noise. In our current setting, rather than directly observing the state, we obtain a high-dimensional observation derived from an injective linear function of the state. Specifically, we assume that $y^{(t)}[i] = Gx^{(t)}[i] + w^{(t)}[i]$, where $G \in \mathbb{R}^{25 \times 4}$ represents the high-dimensional linear mapping. The injective linear mapping matrix $G$ is generated by applying a `thin_svd` operation to a random matrix with values drawn from a normal distribution $\mathcal{N}(0, 1)$. This process ensures injectiveness with a high probability.

For this aforementioned multi-task imitation learning setting, we adopt a scheme in which we gather observations of the form $\{\{y^{(t)}[i], u^{(t)}[i]\}_{i=0}^{N-1}\}_{t=1}^{T}$ from the initial $T$ expert controllers to learn the controller $K^{(T+1)}$. These observations are obtained by following the dynamics:

$$y^{(t)}[i] = (\tilde{A} + \tilde{B}\tilde{K}^{(t)})y[i] + \tilde{B}z^{(t)}[i] + w^{(t)}[i]$$

with $\tilde{A} = GAG^\dagger$, $\tilde{B} = GB$, $\tilde{K}^{(t)} = K^{(t)}G^\dagger$, and process noise $w^{(t)}[i] \sim \mathcal{N}(0, \Sigma_w)$.

The collection of stabilizing LQR controllers $K^{(1)}, K^{(2)}, \ldots, K^{(T+1)}$ is generated by assigning different cost matrices, namely $R = \frac{1}{4}I_{n_u}$ and $Q^{(t)} = \alpha^{(t)}I_{n_x}$, where $\alpha^{(t)} \in$ `logspace`$(0, 3, H)$. These matrices are then utilized to solve the Discrete Algebraic Riccati Equation (DARE): $P^{(t)} = A^\top P^{(t)} A^\top + A^\top P^{(t)} B(B^\top P^{(t)} B + R)^{-1} B^\top P^{(t)} A + Q^{(t)}$, and compute $K^{(t)} = -(B^\top P^{(t)} B + R)^{-1} B^\top P^{(t)} A$, for all $t \in [T+1]$. Moreover, the system matrices $A$ and $B$ are randomly generated, with elements drawn from a uniform distribution. The trajectory length $N = 75$ remains consistent for all tasks. The shared representation is initialized by applying a random rotation to the true representation, denoted as $\Phi_\star = G^\dagger$.

Figure 4 presents a comparative analysis between Algorithm 1 and the vanilla alternating minimization-descent approach (`FedRep` in Collins et al. (2021)) for computing a shared representation across linear quadratic regulators. This shared representation is then utilized to derive the learned controller $K^{(T+1)}$ in a few-shot learning manner. Consistent with our theoretical findings and in alignment with the trends observed in Figures 2-3, Figure 4 demonstrates a substantial reduction in the error between the current representation and the ground truth representation when leveraging

data from multiple tasks, compared to the single-task scenario in Algorithm 1. Furthermore, this figure underscores the significance of de-biasing and whitening the feature data in overcoming the bias barrier introduced by non-iid and non-isotropic data. In contrast, the vanilla alternating descent algorithm fails to address this challenge adequately and yields sub-optimal solutions.