# OpenReview forum: "Sample-Efficient Linear Representation Learning from Non-IID Non-Isotropic Data"
_ICLR.cc/2024/Conference — ICLR 2024 spotlight_

### Official Review · Reviewer_St8o · 2023-10-19

**Soundness:** 3 good
**Presentation:** 4 excellent
**Contribution:** 3 good
**Rating:** 8
**Confidence:** 3

**Summary:**

The authors study the problem of linear representation in a multi-task regression setting. As a starting point, they use an alternating minimization procedure (AMD) developed in prior works on the same problem. They showed empirically that this procedure can fail to learn the correct representation when there is noise in the observations or non-isotropic covariates, even when the different tasks are identical, and gave a theoretical explanation for the sources of error. Based on their analysis, they propose a modification to the alternating minimization procedure (dubbed DFW) which can handle noisy observations and non-isotropic covariates, and experiments confirm the efficacy of their modification.

**Strengths:**

**Clarity of exposition.** The paper is very well written and easy to follow. The authors give extensive interpretation of their results which greatly contributed to my understanding of the paper. The precise relationship to previous work is made explicit, so readers unfamiliar with this sub-field can still parse the paper and understand its contribution easily.

**Intuitive and well-motivated algorithm.** The shortcomings of the base algorithm (AMD) are explained clearly, as are the modifications the authors proposed in DFW, making for an intuitive algorithm. The modifications are simple, easy to implement, and obtain near optimal sample complexity rates.

**Technical contribution.** The authors remove strong technical assumptions found in previous work. They show both theoretically and empirically that these strong assumptions are necessary for AMD to succeed, and are not merely artifacts of previous proofs. Their results are strong both statistically (obtaining optimal sample complexity) and algorithmically (not requiring access to optimization oracles for non-convex problems, which were assumed in some previous works). Their algorithm is also constructed in such a way that data does not need to be shared in its explicit form across tasks, making it attractive when data privacy is a concern. (Remark: It is unclear if the representation _updates_ from each task will still leak private information, but anyway this is not the main focus of the paper.)

**Weaknesses:**

**Theory.** While the assumptions are much weaker than those in related works, some of the assumptions are still very strong. Two in particular stand out.
1. The representation dimension $r$ is required to be at most $\min(d_x, d_y)$, where $d_x$ is the dimension of the covariates and $d_y$ is the dimension of the observations (Section 2, just after equation (1)). In the linear regression setting, this would mean that there must be a one-dimensional representation. This is a very strong assumption. It seems like we should still be able to obtain some benefit if the representation only has a lower dimension than the _covariates_. This would more closely mirror practical settings such as e.g. computer vision, where the data are assumed to belong to a lower-dimensional manifold.
2. Assumption 3.1: It is assumed that the $\beta$-mixing coefficient follows an _exact_ geometric decay, i.e., $\beta^{(t)}(k) = \Gamma^{(t)} \mu^{(t)k}$ for each task $t$. This should place strong restrictions on the possible types of covariate trajectory distributions. It seems like we should expect the results if the decay is _at least_ geometric in nature, i.e., $\beta(k) \leq \Gamma \mu^k$ for some $\mu < 1$.

**Experiments.** The empirical results would be more convincing at showing a fundamental limit on the accuracy for AMD if final accuracy vs. number of tasks was shown at a fixed sample size per task, and showing that this accuracy does not approach 0 as the number of tasks increases. At present, it is just shown for T=25. While DFW does converge in this scenario, in principle, it could just be that DFW has a better sample complexity, but AMD will still eventually converge given enough tasks, albeit at a slower rate. Adding this experiment would strengthen the paper.

A minor point: the title of the OpenReview submission does not match the title on the paper. This should be fixed.

**Questions:**

1. I am curious why the required number of samples $N$ grows (moderately) with the number of tasks $T$. I assume this is to enforce some sort of uniform bound on the random fluctuations across all of the tasks. Can the authors confirm if this intuition is correct?

2. Is there some intuition for why the representation dimension $r$ must be smaller than both the covariate _and_ measurement dimensions? If this is a necessary assumption, can the authors comment on how they would justify this restriction, especially in the linear regression case when $d_y=1$?

3. In Definition 3.1, is there an implicit assumption that the stationary distribution $\nu_\infty$ exists, or are there some conditions imposed on the covariate trajectory distributions which guarantee that a stationary distribution will exist as a consequence?

4. Do the results still hold if the equality for $\beta(k)$ in Assumption 3.1 is replaced with an inequality?

5. It is very interesting that the use of an MLP allows the original AMD algorithm to overcome the fundamental lower bound on the error present when learning a linear representation (even if the sample complexity is much worse than DFW). Is this just because the quantity being measured (validation loss instead of subspace distance) is different, or would AMD with a linear representation fail to converge to 0 validation loss in this setting? If this is particular to the MLP representation, do the authors have any intuition for why this might be the case?

---

> ### Author Response · Authors · 2023-11-18
> **Author response (1/2)**
>
> We thank the reviewer for their detailed comments. To address the remaining questions/concerns:
>
> - The most updated title of our paper is the one on OpenReview: “Sample-Efficient Linear Representation Learning from Non-IID Non-Isotropic Data”. We apologize for the confusion.
>
> - The $r \leq \min\{d_y, d_x\}$ is a typo; as one should expect, we only require $r \leq d_x$ for our analysis and can accommodate arbitrary $d_y$, e.g. $d_y = 1 \ll d_x$ for scalar-output linear regression, or $d_y \approx d_x$ in linear system identification (Appendix B.1). We thank the reviewer for identifying this important oversight.
>
> - The beta-mixing coefficient only needs to be upper bounded by geometric decay. This will be fixed in the revision.
>
> - **Regarding the requested experiment for alternating minimization-descent (AMD)**: we took the same experimental parameters for linear representations as described in Section 4.1 (corresponding to Fig 1a), except varying the number of total tasks $T \in \{1, 5, 10, 25, 50, 100\}$. For each $T$, we run alternating minimization-descent for a single run of 5000 iterations ($\gg 100$ iterations shown in Fig 1a and more than in the MLP experiment) and record the subspace distances. For comparison, we run DFW with a single task $T=1$. This generates the following figure ([anonymous google drive link](https://drive.google.com/file/d/1ztBpxcl0MTcoupD8L_DZ1gsRENrCQPGr/view?usp=sharing)). The plot demonstrates that: 1. increasing the number of tasks *cannot push AMD past a (large) threshold*, only serving to reduce the variance, precisely as our theory predicts, 2. *AMD does not encounter a feature learning phase* as does an MLP (Figure 1b) that drives the distance to $0$, even after many iterations. In comparison, DFW even for a single task quickly converges to $0$, confirming that the suboptimality of AMD is due to a fundamental bias, rather than the noise level of the problem.
>
> - **Regarding the moderate growth of $N$ with respect to tasks $T$**: this is a question with some subtlety. There are two sources to the $\log(T)$ dependence–one that arises in the analysis of the iid setting, and the other from $\beta$-mixing.
>    - In the iid setting, as the reviewer suspects, the $\log(T)$ dependence comes from an eventual union bound (a.k.a. uniform control) over the noise processes. In particular, to bound the largest singular value of the average of self-normalized martingale (SNM) noise across tasks $T$, we appeal to a Matrix Hoeffding inequality (Lemma A.4). However, like the traditional Hoeffding inequality, this requires boundedness in the psd order of each summand (i.e. each task’s SNM). Since we have concentration inequalities for individual SNMs (Propositions A.1 and A.2), a simple way to simulate boundedness is to condition on the high-probability boundedness event of each task’s SNM. However, conditioning simultaneously on each task’s boundedness equates to a union bound, which leads to a $\log(T)$ factor when inverted for a desired failure probability $\delta$. The $\log(T)$ factor cannot be avoided when going through the route of boundedness/truncation even though SNMs are independent between different tasks (see e.g. subgaussian maximal inequalities). *However*, we conjecture that this $\log(T)$ factor is only technical, and *may be removable in the iid setting*. This requires going from a (matrix) Hoeffding to a Bernstein-type inequality, which in turn requires control of the higher-order moments of the SNM. As far as we can tell, this would require highly non-trivial, novel analysis of self-normalized martingales, and importantly, **the mild savings will be washed away (order-wise) in the $\beta$-mixing setting**, where they arise for a fundamental reason. (continued in next comment)

---

> > ### Author Response · Authors · 2023-11-18
> > **Author Response (2/2)**
> >
> > - (continued from previous comment)
> >    - For $\beta$-mixing processes, standard tools (e.g. Lemma A.3) relate the dependent process to the iid resampled process, at the cost of an additive factor to the probability of the event that depends on the mixing coefficients. Again, since we pay this additive factor for each task’s process (or equivalently, viewing our data as a single $\beta$-mixing sequence of length $TN$), we pay a $\log(T)$ (and $\log(N)$) factor when inverting for a desired failure probability $\delta$. This task-dependence is likely fundamental for the following reason: our algorithm critically ensures that the gradient noise (for DFW, in the form of SNMs) are zero-mean matrices in the iid setting, such that averaging across tasks causes the gradient noise matrix to converge to $0$ as $T$ increases. However, when the gradient noise is biased, then as $T \to \infty$, averaging simply concentrates to the bias. For mixing sequences, the gradient noise is no longer precisely zero-mean due to dependence across covariates. For simplicity, let us assume identical data distributions across tasks. For a *fixed* per-task data budget $N$, as we grow $T \to \infty$, averaging gradient noise across tasks will eventually converge to the bias of the task-wise SNM (which only depends on $N$). Therefore, to ensure the gradient noise decays to $0$ at rate $\tilde {\mathcal O}(1/NT)$ for any given $T$, the per-task requirement on $N$ necessarily depends on $T$--in the geometric mixing case this amounts to the mild $\log(T)$ factor.
> >
> > - **Regarding assuming an exact stationary distribution**: we use exact stationarity for notational and expositional simplicity. Going from exact stationarity to asymptotic stationarity (see [1]) is rather painless in our setting, and simply requires modifying the $\beta$-mixing definition/assumption accordingly, since the technical keystone Lemma A.3 is in fact originally stated for asymptotically stationary sequences. Regarding what kinds of covariate sequences exhibit stationarity, (asymptotic) stationarity arises naturally in our exemplar applications in linear dynamical systems; we refer to Appendix B for a detailed discussion. However, regression over ($\beta$-mixing, asymptotically) stationary covariates has a storied history far beyond our applications of choice, e.g. over Markov chains (cf. [2]), ARMA processes [3], certain GARCH models [4] etc.
> >
> > - **Regarding the MLP representation experiments**: as suggested by the additional experiment described earlier, a linear parameterization in conjunction with vanilla alternating minimization-descent does not converge no matter how much data is provided. The ability of the MLP to break past this barrier likely extends beyond the measured quantity (validation loss), since the validation set is sampled from a distinct task that is not used in training. In terms of why the MLP parameterization is eventually able to descend to optimality: we do not claim to be experts on the theory of neural network optimization, but a simple hypothesis is that the MLP is expressive enough to find a non-linear parameterization that leads to a path that escapes the local minima, which seems to match the folk understanding that SGD tends to find globally near-optimal solutions for many non-convex objectives in deep learning (e.g. [5]). We generated a figure ([anonymized google drive link](https://drive.google.com/file/d/1h7zpLqaLmCVqPSKBLCWcOJoHHub8ffgB/view?usp=sharing)) that shows the norm of the MLP gradient evolving in parallel to Figure 1b, which demonstrates that SGD does indeed approach a local minima before finding a path out toward the global minima, reflected by the noticeable bump in the gradient norm mirroring the generalization phase in Figure 1b.
> >
> > [1] Vitaly Kuznetsov and Mehryar Mohri. “Generalization bounds for non-stationary mixing processes”.
> >
> > [2] Guy Bresler, Prateek Jain, Dheeraj Nagaraj, Praneeth Netrapalli, Xian Wu. “Least Squares Regression with Markovian Data: Fundamental Limits and Algorithms”
> >
> > [3] Abdelkader Mokkadem. “Mixing properties of arma processes”
> >
> > [4] Marine Carrasco and Xiaohong Chen. “Mixing and moment properties of various garch and stochastic volatility models”
> >
> > [5] Simon Du, Jason Lee, Haochuan Li, Liwei Wang, Xiyu Zhai. “Gradient Descent Finds Global Minima of Deep Neural Networks”

---

> > > ### Comment · Reviewer_St8o · 2023-11-20
> > >
> > > Thank you to the authors for their very thorough response. All of my concerns have been fully addressed, and the technical discussions provided by the authors (specifically on the log(T) dependence for N) provided even more insight into the results. The other reviewers raised some interesting questions, and it seems that the authors have resolved these as well. I have increased my score accordingly.

---

### Official Review · Reviewer_ZRaH · 2023-10-27

**Soundness:** 3 good
**Presentation:** 3 good
**Contribution:** 3 good
**Rating:** 8
**Confidence:** 4

**Summary:**

This paper proposes an algorithm about learning the representation is a linear connection between feature and labels. The algorithm is based on gradient descent and QR decomposition on the iterates. The paper further proves a bound about the sample complexity and error of the algorithm, which is optimal in terms of problem parameters (degree of freedom). Numerical experiments validates the performance of the algorithm.

**Strengths:**

This paper proposes A practical and simple algorithm, and the theories as well as the math proof of the sample complexity (per batch and in total) and error are solid in terms of the degree of the freedom. The logic and the writing is clear.
Especially, Remark 3.2 is great where we can see that the lower bound of $N$ makes sense. Some other papers, although claiming optimality with respect to total samples $NT$, there is a strong assumption on lower bound $N$ that makes them trival, e.g., Du et al.

**Weaknesses:**

On the other hand, does Tripuraneni et al. work when $N = O(1)$? This paper assumes $N = \Omega(r)$ so there is still a gap from the optimum. The result in this paper is still good because $r$ it's a small number in low rank setting which we are interested in, and it is already better than the papers listed in Remark 3.2. But it would be great to propose why this paper cannot achieve $N = O(1)$.

Since this paper discusses general feature covariances, it would be great to talk more about the impact of the spectrum of the covariance matrix. There are a few papers about how the feature and operator covariances’ spectrums show up in the bound, and how the "aligned" covariances help learning, for example,

Wu and Xu, On the Optimal Weighted $\ell_2$ Regularization in Overparameterized Linear Regression

And a few relevant ones.

It would be great to have a notation table, either in main text or appendix, because there are many different notations/definitions.

**Questions:**

No more questions.

---

> ### Author Response · Authors · 2023-11-18
> **Author Response**
>
> We thank the reviewer for their evaluation. To address the remaining questions:
>
> - We are introducing a notation table to our revised appendix.
>
> - **Regarding the burn-in requirement $\Omega(r)$**: the requirement in Tripuraneni et al. (and similar papers) is indeed $\Omega(1)$. However, we strongly emphasize that the proposed algorithm in that paper critically relies on the fact that the covariates across **all tasks** are iid and isotropic. From an algorithmic perspective, their proposed method-of-moments estimator is designed with the implicit foreknowledge that $\mathbb E[x x^\top] = I$, and is only guaranteed to be consistent then. On the other hand, their proposed lower bound on the subspace distance, by assuming all covariates are iid $\mathcal N(0,I)$, in fact does not even require $N > 0$ for any given task, as long as the total samples $NT$ is large enough. However, this quirk is inextricably tied to the iid covariate assumption, which is reflected in the proof of the result (Lemma 22 of their appendix). Intuitively, we should not expect $N = \Theta(1)$ to be possible, in the sense that allowing the covariances across each task to differ permits the design of weights $F^{(t)}$ and covariances $\Sigma_x^{(t)}$ that are arbitrarily ill-conditioned such that distinguishing $\Phi_\star$ and a perturbed $\Phi’_\star$ from sampled-data is provably hard unless a sufficient per-task burn-in on $N$ (e.g. proportional to the row-rank of $\Phi$) is satisfied. Formalizing this lower bound, as well as extracting the task-diversity/task-coverage assumptions of the task-wise covariances $\Sigma_x^{(t)}$ that do not manifest in the task-wise iid setting is ongoing work.
>
> - We agree that **random features in addition to random covariate design** is an interesting consideration. We note that when we set the weights $F^{(t)}$ to be random (e.g. Gaussian random matrix), keeping $\Phi_\star$ as deterministic (otherwise the low-rank representation learning problem changes), our analysis actually goes through *mutatis mutandis*: in particular, instead of deterministic task diversity parameters $\lambda_{\min}^{\mathbf F}, \lambda_{\max}^{\mathbf F}$ (Definition 3.2), we can replace them with high-probability variants, carrying out the rest of the analysis as written and simply accruing the additional failure probability in the final bound. **Regarding the alignment of covariances**, we note some subtlety in Theorem 3.1 that was washed out for simplicity: the dependence on task-specific quantities, such as the smallest covariance eigenvalue $\lambda_{\min}(\Sigma^{(t)}_x)$, are actually averaged across tasks in a certain way, where they currently appear as a max over tasks $t$ for simplicity. After inverting for sample complexity bounds as in Corollary 3.1, these task "coverage" / "overlap" quantities naturally manifest in the bound, which may be seen as notions of how the alignment and/or coverage of source task covariances affect learning. Lastly, a **theory for an overparameterized setting** is quite interesting, and seems to be getting concurrent attention; however, the data / feature assumptions are necessarily quite distinct from ours, and thus we cannot claim to easily extrapolate the trends in our results there.

---

### Official Review · Reviewer_6Cex · 2023-10-28

**Soundness:** 2 fair
**Presentation:** 3 good
**Contribution:** 3 good
**Rating:** 6
**Confidence:** 3

**Summary:**

This paper points out a failed example of traditional algorithms in handling non-isotropic data. To overcome this issue, it proposes an algorithm called De-bias & Feature-Whiten (DFW) for multi-task linear representation learning from non-iid and non-isotropic data. DFW provably recovers the optimal shared representation at a rate that scales favorably with the number of tasks and data samples per task. Numerical verification is also provided to validate the proposed algorithms.

**Strengths:**

Regarding the originality, few meta- federated- learning papers are working on non-iid settings. So this paper has its own novelty.

The paper is also well-structured and clearly states the necessary backgrounds, though some technical details should be further extended.

The example on the non-IID non-isotropic data provides a clear motivation of proposing a new algorithm to overcome this issue. It indicates parts of significance of this work.

**Weaknesses:**

The title "META-LEARNING OPERATORS TO OPTIMALITY FROM MULTI-TASK NON-IID DATA" is so vague. It is really hard to understand what this paper studies from the title. It should indicate that the goal is to learn the shared parameter $\Phi$.

The failed example given in Section 3.1 serves as the main motivation of introducing new algorithms. However, these two crucial issues in this example are not really resolved. I am concerned if the de-bias and feature-whiten steps could really resolve these issues. I put more comments in the next section.

**Questions:**

1. First about clarifying the key idea de-bias and feature-whiten methods. In Section 3.2, it says that $\hat{F}^{(t)}$ is computed on independent data. It is not clear to me why $\hat{F}^{(t)}$ is independent from $X^{(t)}$.  To my understanding, for example, the Partition trajectories step (Line 5, Algorithm 1) splits the dataset $N$ to $N_{1}=\\{x_1, x_2, \dots, x_n \\}$ and $N_{2}=\\{x_{n+1}, x_{n+2}, \dots, x_{n+N} \\}$. But they come from the same $\beta$-mixing stochastic process, will they become independent?

2. What is "the aforementioned batching strategy" mentioned in Section 3.2 right after Eq.(5)? It seems that there is no batching strategy mentioned before.

3. The proof for the non-iid case simply says after taking the "blocking technique on each trajectory", everything is same as the iid case. First, what is the "blocking technique on each trajectory"? Has this technique been introduced before?

4. Then regarding the proof for the non-iid case, I am mainly concerned if the iid case could be simply immigrated to the non-iid case. For example, on page 16 of the supplimentary material, it says "We observe that since $\hat{F}(t)$ is by construction independent of ... " and obtains $$E[FWX\Sigma^{-1}]=E[F]E[W]E[X\Sigma^{-1}]$$
This equation won't hold for the non-iid case. It is because $F$ here is evaluated using a part of the process $\{x\}$ and $\Sigma$ is estimated using another part of the same process.

---

> ### Author Response · Authors · 2023-11-18
> **Author Response**
>
> We thank the reviewer for their detailed comments. To address their questions:
>
> - The title in the submitted pdf is in error. The most updated version is the one seen on the OpenReview page “Sample-Efficient Linear Representation Learning from Non-IID Non-Isotropic Data“, which clarifies the goal of learning a shared representation.
>
> - In general, **regarding the possible correlation due to sampling from the same $\beta$-mixing process**, our main technical workhorse is Lemma A.3, which is a standard tool used in the analysis of mixing processes. In short, the expected value of a bounded measurable function of a given $\beta$-mixing process and the same function on the iid version of the process (where each covariate $x_i$, $i=1,\dots,$ is sampled from its population distribution), can be bounded in terms of the mixing function $\beta(\cdot)$. In particular, if said function is an indicator of an event of interest, e.g. the complement of the descent guarantee Eq (11) in Theorem 3.1, then this implies the probability of the event occurring on the mixing process versus the independent version is the same up to an additive factor that depends on the mixing function. Therefore, the iid results hold for the mixing case, albeit with a smaller effective sample size in order to achieve the same failure rate $\delta$. For geometric mixing processes, this essentially amounts to reducing $N \to N/\log(N)$ samples. This addresses a couple of questions:
>    - **The discussion of the “blocking” technique** is located around Definition A.1 ($\beta$-mixing) and Lemma A.3. We will make references to blocking clearer in our revision.
>    - **Regarding independence of $X^{(t)}$ and $\hat F^{(t)}$: in the mixing case**, they are indeed possibly not independent. However, as discussed, their behavior can be bounded by the analysis of the iid case, paying the appropriate costs for dependency in the mixing coefficients. Similarly, the decomposition of $F, W, X\Sigma^{-1}$ need only hold for the iid setting.
>
> - **The “aforementioned batching strategy” mentioned after Eq (5)** refers to the discussion just preceding Eq (5): “each agent computes the least squares weights $\hat F^{(t)}$ and the representation update on independent batches of data, e.g. disjoint subsets of trajectories”. We will adjust the wording in the revision, as we indeed do not discuss a concrete strategy until the subsequent paragraphs.
>
> In light of this discussion, we hope that this answers the reviewer’s concerns about whether our proposed algorithm and analyses fully overcome the identified non-isotropy and non-iid issues, as far as the $\beta$-mixing assumption holds.

---

> > ### Comment · Reviewer_6Cex · 2023-11-18
> >
> > Thanks for the clarification! Now I could understand the proof of the non-iid case. Since my concerns are addressed, I will increase my score accordingly.

---

### Official Review · Reviewer_4V27 · 2023-10-30

**Soundness:** 3 good
**Presentation:** 4 excellent
**Contribution:** 3 good
**Rating:** 8
**Confidence:** 3

**Summary:**

In "Sample-Efficient Linear Representation Learning from Non-IID Non-Isotropic Data" proposes a scheme and statistical guarantees to problems stemming from multi-task learning. In this setting, prior works focused on i.i.d. and isotropic data while the proposed work allows for non-i.i.d. and non-isotropic data. In order to design the scheme and provide with statistical guarantees, in which learning all tasks jointly implies a statistical gain, the authors modify a proposed scheme for the i.i.d. and isotropic data by including mini-batches and whitening.

The obtained results are what is expected in terms of statistical precision, given the total number of samples, tasks and problem dimension.

**Strengths:**

The paper is overall well written, and I think the result is good enough. While a criticism can be put forth in that it combines existing known techniques to establish the final result, it is not necessarily obvious that the combination yields the desired statistical result.

**Weaknesses:**

I am overall happy with the paper. I think the authors did a good job at presenting their work. I mainly have two questions/weaknesess. The result provided by the authors requires a minimum number of samples under which contraction upto a ball of the alignment of the estimated and optimal subspace are. Is there any sense to how tight this bound is from an information theoretical sense, i.e. the scaling with gamma, mu, etc?

Second, in corollary 3.1. the authors establish the existence of a partition of independent batches that guarantees this result. Can it be guaranteed that such partition is found in practice?

**Questions:**

- While both are valid, the title in the pdf file and the title given within the open review system do not match.

---

> ### Author Response · Authors · 2023-11-18
> **Author Response**
>
> We thank the reviewer for their comments. To address the reviewer’s questions:
>
> - The most updated title of our paper is the one on OpenReview: “Sample-Efficient Linear Representation Learning from Non-IID Non-Isotropic Data”. We apologize for the confusion.
>
> - **Regarding Corollary 3.1**, the requisite partitions are actually very simple to construct in practice. In short, given the assumptions of Theorem 3.1, we are guaranteed the distance-to-optimality of the next iterate is decomposed into a contraction of the previous iterate’s distance and an additive noise term. Therefore, the idea that yields Corollary 3.1 is to simply partition a given offline dataset into exponentially growing chunks, such that the noise term is sufficiently small with respect to the contraction. Namely, if $d_{t+1} \leq \rho \cdot d_t + \frac{\sigma}{\sqrt{n}}$, then any schedule for $n$ that ensures $d_{t+1} \leq \rho’ \cdot d_t$, $\rho’ < 1$, $t =0,1,\dots$ suffices. For an offline dataset of size $N$, an exponential growing partition can be accommodated $K = O(\log N)$ times, which yields $d_K \lesssim C(\rho’) \frac{\sigma}{\sqrt{N}}$, hence approximating “ERM-like” rates of $\approx \frac{\sigma}{\sqrt{N}}$. We note that this is a straightforward adaptation of the standard “doubling trick” in online learning, and discussed in further detail in Lemma A.11. Any exponential schedule can be used in principle, as long as contraction is enforced (visualized in practice, for example, via a validation loss), without hurting the rate of the final bound.
>
> - **Regarding tightness of dependence on various problem quantities**. Starting with our per-task sample requirement $N \geq \Omega(r + \sigma_w^2 \max\{d_y, r\})$, we do not provide a formal minimax lower bound in this paper, but we have reasons to believe it cannot be significantly improved, e.g. $N = \mathcal O(1)$, in general. For $d_y = 1$, the lower bound of $N = \mathcal O(1)$ provided by Tripuraneni et al. critically depends on the covariates being iid $\mathcal N(0,I)$ across all tasks. Allowing covariances to change across tasks opens the door to arbitrarily ill-conditioned interactions between weights $F^{(t)}$ and covariances $\Sigma_x^{(t)}$ which cannot happen when $\Sigma_x^{(t)} = I$. Formalizing the resulting local-minimax lower bound is ongoing work. Whether the mixing coefficients $\Gamma, \mu$ in our final bounds is optimal is a challenging question. Indeed, recent work [1] establishes in the general case that a final error bound on least squares regression scaling as $\frac{\tau_{\mathrm{mix}}}{N}$ (for $N$ total data points) is minimax-optimal, where $\tau_{\mathrm{mix}}$ is proportional to the dependence on $\log(N), \Gamma, \mu$ in our bounds, at least in the Markov chain setting. However, in parallel, even more recent work [2] demonstrates that under a related notion of mixing, the dependence on the mixing time becomes lower order after sufficiently large burn-in on $N$, after which the iid regression rate $\mathcal O(1/N)$ is recovered. Regardless, these works are specific to estimators such as the empirical risk minimizer which are distinct from ours, not to mention the additional bilinear structure on our multi-task operators. In short, fully determining whether the dependence on mixing coefficients in our bounds is optimal is non-trivial, though we hypothesize that it is not, thanks to the certain amenable features of our model, such as realizability, subgaussian covariates etc.
>
>
> [1] Guy Bresler, Prateek Jain, Dheeraj Nagaraj, Praneeth Netrapalli, Xian Wu. “Least Squares Regression with Markovian Data: Fundamental Limits and Algorithms”
>
> [2] Ingvar Ziemann and Stephen Tu. “Learning with little mixing”

---

> > ### Comment · Reviewer_4V27 · 2023-11-19
> >
> > Thank you for your response! I am happy to increase my score.

---

### Meta-Review · Area_Chair_PYiD · 2023-12-14

**Metareview:**

Authors study a setting of recovering linear operators from noisy vector measurements, which is based on gradient descent (GD) and QR decomposition on the  iterates. The authors establish bounds on the sample complexity and error of the algorithm, which is optimal in terms of problem parameters. Authors provide numerical experiments to demonstrate the performance of the proposed algorithm.

This paper was reviewed by 4 reviewers and received the following Rating/Confidence scores: 8/3, 8/3, 8/4, 6/3.

I think the paper is overall interesting and should be included in ICLR. The authors should carefully go over and clarify all reviewers' questions/concerns.

**Justification For Why Not Higher Score:**

n/a

**Justification For Why Not Lower Score:**

n/a

---

### Decision · Program_Chairs · 2024-01-16

Accept (spotlight)